# On the Adversarial Robustness of Benjamini Hochberg

**Louis L Chen**[*]
Operations Research Department
Naval Postgraduate School
Monterey, CA 93943
louis.chen@nps.edu

**Roberto Szechtman**
Operations Research Department
Naval Postgraduate School
Monterey, CA 93943
rszechtm@nps.edu

**Matan Seri**
Operations Research Department
Naval Postgraduate School
Monterey, CA 93943
matan.seri@gmail.com

## Abstract

The Benjamini-Hochberg (BH) procedure is widely used to control the false detection rate (FDR) in multiple testing. Applications of this control abound in drug discovery, forensics, anomaly detection, and, in particular, machine learning, ranging from nonparametric outlier detection to out-of-distribution detection and one-class classification methods. Considering this control could be relied upon in critical safety/security contexts, we investigate its adversarial robustness. More precisely, we study under what conditions BH does and does not exhibit adversarial robustness, we present a class of simple and easily implementable adversarial test-perturbation algorithms, and we perform computational experiments. With our algorithms, we demonstrate that there are conditions under which BH's control can be significantly broken with relatively few (even just one) test score perturbation(s), and provide non-asymptotic guarantees on the expected adversarial-adjustment to FDR. Our technical analysis involves a combinatorial reframing of the BH procedure as a "balls into bins" process, and drawing a connection to generalized ballot problems to facilitate an information-theoretic approach for deriving non-asymptotic lower bounds.

## 1 Introduction

Multiple testing has broad applications in drug discovery, forensics, candidate screening, anomaly detection, and in particular, machine learning. Indeed, recent works [5, 18, 28, 32], in nonparametric outlier detection, *out-of-distribution detection* (OOD), and one-class classification have all adopted multiple testing methodology in developing principled decision rules with statistical guarantees. In fact, the Benjamini-Hochberg (BH) multiple testing procedure, widely used to control the *false detection rate* (FDR), is either used or modified in all these recent methods. Considering this FDR control could be relied upon in some critical (safety/security) contexts, for which false positives incur costs, we investigate its adversarial robustness.

Adversarial corruption presents a challenge to statistical methodology, and is a modern-day concern due to not only the ease with which high volumes of data can now be accessed/processed but also the increasingly widespread use of statistical procedures. This threat poses vulnerabilities to machine learning tasks like OOD, which would aim to fortify security systems like fraud detection [5].

---

[*]website: https://louislchen.github.io/

38th Conference on Neural Information Processing Systems (NeurIPS 2024).

Manipulation of data and experimental results are common means by which incorrect conclusions can be reached. Worse, strategic perturbation can dramatically decrease the fidelity of the models and methods used. A burgeoning field of adversarial corruption has gained traction in recent years to meet this concern, most notably in the area of (deep) machine learning; see, for example, [27, 16, 20]. In this work we address adversarial corruption in hypothesis testing, specifically in the large-scale context in which the primary focus is on the aggregate metric: FDR.

BH [6] is one of the most widely used multiple testing procedures, which upon input of a collection of p-values, outputs a rejection region ensuring that the FDR is no greater than a user-defined threshold $q \in (0, 1)$. This control of FDR holds under independently generated p-values – as well as some restricted forms of dependency like *positive regression dependent on a subset* (PRDS) [7] – but it generally holds without strong assumptions on the alternative distributions. This degree of distributional robustness, however, could be said to come at the cost of adversarial robustness, as we show in this work.

## 1.1 Literature Review

Although OOD methods [23, 12, 21, 22] are often complex and not always supported by statistical guarantees, conformal inference has made possible the use of one-class classifiers to generate conformal p-values for which OOD can now be conducted via multiple testing. This has led to the adoption of the BH procedure in OOD. Indeed, [5] leverages the FDR control afforded by BH over conformal p-values (shown to be PRDS) to test for outliers. More precisely, given a test set of observations for which we wish to identify as inliers or outliers (out of distribution), a conformal p-value is generated for each observation, which is then processed by BH to decide which are likely outliers. We refer the reader to [18, 28] for other recent works along this vein.

In recent years, concerns have risen over the possibility of adversarial manipulation of statistical methodologies. This manipulation commonly occurs at the level of data collection and training, often invalidating the assumptions made regarding how data is drawn, but it can also occur at test time. There is a growing literature on *adversarial robustness*, which is concerned with securing statistical methods like (deep) machine learning [27, 16, 20], linear regression [8], M-estimation [9], and online learning [26, 17, 1]. In particular, [13] considers contamination models that incorporate (adaptive) adversarial perturbation of up to an $\epsilon-$ fraction of drawn data. Indeed, we adopt this modeling in our own study - see (c-Perturb). As well, a similar concept to the notion of adversarial robustness that we adopt in this paper is one the literature refers to as *perturbation resilience*. Generally speaking, a problem instance is called $\alpha-$ perturbation resilient when despite a degree (parameterized by $\alpha$) of perturbation to the instance, the optimal solution does not change. First introduced in [10] for combinatorial optimization (in particular, MAX-CUT), the concept has since also inspired research into devising resilient unsupervised learning, particularly in clustering [4, 3, 2].

With respect to the hypothesis testing literature, there are recent adversarial robust studies focused on simple [19] and sequential hypothesis testing [11] from a game theoretic perspective, in which protection of statistical power, risk, or sample size from corruption is of chief concern. Complementing the adversarial robust perspective are several distributionally robust studies, in which the data-generating distribution is known only to lie in a parametric family. Recent works include [11, 24, 15] which focus on test risk in single and sequential hypothesis testing settings employing uncertainty sets of distributions of fixed distance (e.g. Wasserstein, phi-divergence) for the null and alternative hypotheses. In contrast to these works, this paper is focused on FDR, not individual test risk. Furthermore, distributional robustness is not equivalent to the perturbation-robustness that this paper and other adversarial robust studies seek in general. Indeed, [30] shows that the BH procedure's FDR control exhibits a distributional robustness to possible dependence between null and non-null hypotheses. On the other hand, our work would illustrate that, distributional robustness aside, BH can lack adversarial robustness.

## 1.2 Preliminaries

Let $\mathcal{N} := \{1, \ldots, N\}$, where $N \in \mathbb{Z}_+$, be a finite set for which each member $i \in \mathcal{N}$ denotes a binary hypothesis test deciding between a null and alternative hypothesis. Further, there exists a partitioning, $\mathcal{N} = \mathcal{H}_0 \uplus \mathcal{H}_1$, such that the correct decision for test $i \in \mathcal{N}$ is either null if $i \in \mathcal{H}_0$, or alternative if $i \in \mathcal{H}_1$. Here, $\mathcal{H}_0$ and $\mathcal{H}_1$ are the (unknown) sets of null and alternative test indices, respectively;

consequently, for any test $i$, the correct decision (i.e. set membership) is unknown to the decision maker. In fact, while the number of tests $N$ is known (and large, on the order of thousands), neither $N_0 := |\mathcal{H}_0|$ nor $\pi_0 := \frac{N_0}{N}$ is known to the decision maker, although $\pi_0 \geq 0.90$ "is reasonable in most large-scale testing situations" - ([14], p. 285).

For each test $i \in \mathcal{N}$, p-value $p_i \in [0, 1]$ is randomly generated (independent of all other $p_j$, $j \neq i$), which we model as a draw from either $U(0, 1)$ when $i \in \mathcal{H}_0$ ($p_i$ then referred to as a *null p-value*) or some alternative distribution $\mathbb{P}_i^1$ on $[0, 1]$ when $i \in \mathcal{H}_1$ ($p_i$ then referred to as an *alternative p-value*). A multiple-testing algorithm $\mathcal{A}$ takes as input a randomly generated collection of p-values $p = \{p_i\}_{i \in \mathcal{N}}$ and outputs for each test $i$ a determination $\mathcal{A}(i) \in \{0, 1\}$ with $\mathcal{A}(i) = 1$ iff the determination is to reject the null hypothesis (i.e., claim $i \in \mathcal{H}_1$), or sometimes referred to as "make a discovery" for the $i$-th test. With $a_p := \sum_{i \in \mathcal{H}_0} \mathcal{A}(i)$ denoting the number of false discoveries, and $R_p := |\mathcal{A}^{-1}(1)|$ denoting the number of rejections/discoveries made, we refer to $FDP[\mathcal{A}; p] := \frac{a_p}{R_p \vee 1}$ as the false detection proportion summarizing $\mathcal{A}$'s decisions on p-values $p$, where $x \vee y$ is shorthand for $\max(x, y)$ for any $x, y \in \mathbb{R}$. We refer to its expectation with respect to the random generation of $p$ as the *false detection rate*, $FDR(\mathcal{A}) := \mathbb{E}_p FDP[\mathcal{A}; p]$.

In this work, we focus on the Benjamini Hochberg procedure, a widely-used multiple-testing algorithm.

## 1.3   The Benjamini Hochberg (BH) Procedure

Given a collection of p-values $p = \{p_i\}_{i \in \mathcal{N}}$ and desired control level $q \in (0, 1)$ to bound FDR, the BH procedure $BH_q$ operates as follows:

1. The p-values are sorted in increasing order, $p_{(1)} \leq p_{(2)} \leq \ldots \leq p_{(N)}$.

2. The index $i_{\max} := \max\left\{i \in [0, N]_{\mathbb{Z}} : p_{(i)} \leq i \frac{q}{N}\right\}$ is identified, with $p_{(0)} := 0$.

3. Reject the tests corresponding to the smallest $i_{\max}$ p-values: $p_{(1)}, p_{(2)}, \ldots, p_{(i_{\max})}$.

$BH_q$ provides provable FDR control at the level of $q$ without any assumptions on the alternative distributions $\{\mathbb{P}_i^1\}_{i \in \mathcal{H}_1}$.

**Lemma 1.1** (Theorem 4.1 from [14]). *If every null p-value is super-uniform, equiv., $p_i \sim \mathbb{P}_i^0 \succcurlyeq U(0, 1)$ for all $i \in \mathcal{H}_0$, and the collection is jointly independent, then regardless of the collection of alternative distributions $\{\mathbb{P}_i^1\}_{i \in \mathcal{H}_1}$,*

$$FDR(BH_q) = \pi_0 q \leq q, \quad \forall q \in (0, 1). \tag{1}$$

*Remark* 1.2. In fact, the assumption of joint independence of the null p-values can be relaxed to a form of dependency known as PRDS - see [7] for this generalization of Lemma 1.1.

## 1.4   The Adversary and the c-Perturbation Problem

We model an (*omniscient*) adversary with knowledge of $\mathcal{H}_0$, $\mathcal{H}_1$, and that knows the decision maker's choice of control level $q$. The adversary receives the p-values $p = (p_i)_{i=1}^N$ *after* they are generated but *before* they are received by the decision maker, or before test time. Given the ability to perturb $c \geq 1$ p-values, the adversary solves

$$\max_{p': \|p - p'\|_0 \leq c} FDP[BH_q; p'], \tag{c-Perturb}$$

where $p'$ denotes the p-values derived from an adjusted collection of p-values $p' = (p_i')_{i=1}^N$. In words, the adversary finds the perturbation of at most $c$−many p-values before the execution of $BH_q$ so as to maximize the *adversarially-adjusted* false detection proportion $FDP[BH_q, p']$. Perturbation of p-values is implicitly the result of data perturbation, and we refer to both Remark 4.6 and Section 5.2 for examples and experiments involving direct perturbation of data.

While we assume omniscience for the adversary throughout, we will briefly address modifications in analysis for an *oblivious* adversary that has no knowledge of $\mathcal{H}_0$, nor $\mathcal{H}_1$. Indeed, our algorithms to be presented can be modified naturally for implementation by an oblivious adversary (see comments in Section 3); further, there is nearly equivalent performance when $\pi_0$ is large, as is typical. Hence, it is for the sake of brevity that we omit explicit analysis of the oblivious adversary.

## 1.5 Main Results

Intuitively, the effect of a small number of p-value perturbations becomes insignificant in settings where a large number of tests are rejected (see Theorem 3.2). This happens, for instance, when either the number of tests $N$, or the control level $q$, or the distance (e.g. KL-divergence) between the null and alternative distributions, is large. For this reason, we focus on results that are non-asymptotic in the number of tests $N$.

In Section 3, we present the algorithm INCREASE-c that uses strategic increases to $c$ null p-values to induce expansion of the BH rejection region. We also present an efficient, optimal algorithm MOVE-1 (Appendix Section 7.2) for the adversary's maximization of FDR with at most one (i.e. $c = 1$) p-value perturbation.

In Section 4 we discuss the adversarial robustness of BH through study of the adversarial adjustments to FDR by the INCREASE-c algorithms, revealing where its control is and isn't adversarially robust.

In Section 5 we provide accompanying numerical experiments on i.i.d. as well as PRDS p-values.

## 2 BH as Balls into Bins

In this section we will establish important notation for the discussions to follow. We reduce the real line to a collection of $N + 1$ "bins". $N_0$ and $N_1$ balls will each be assigned to one of these bins independently of each other, from discrete distributions that are specified in the next subsection. The main motivation for this reduction is to facilitate discussion of effective perturbations in Section 3, and for the technical analysis in Section 4.

### 2.1 The Balls into Bins System

We partition the segment $[0, 1]$ into $N$ equiprobable segments that will be referred to as *bins*. We define the $i$-th bin $B_i := \left\{ p \in \mathbb{R} : (i-1)\frac{q}{N} \leq p < i\frac{q}{N} \right\}$, for $i = 1, \ldots, N$. What remains forms bin $N+1$, i.e., $B_{N+1} := \left\{ p \in \mathbb{R} : 1 \geq p \geq 1 - q \right\}$. For shorthand, we write $B_{1:i} := \cup_{l=1}^{i} B_l = \left\{ p \in \mathbb{R} : 0 \leq p < i\frac{q}{N} \right\}$, Finally, we write $B_i^0 := |B_i \cap \{p_j\}_{j \in \mathcal{H}_0}|$ and $B_i^{\mathcal{N}} := |B_i \cap \{p_j\}_{j \in \mathcal{N}}|$ for bin $i$'s, respectively, *null-load* and *total load*. The *alternative-load* $B_i^1$ is defined analogously, as are $B_{1:i}^0$, $B_{1:i}^{\mathcal{N}}$, and $B_{1:i}^1$.

**Rejection Count:** Borrowing terminology from the classic *balls into bins* problem of probability theory [29], this framework facilitates a re-interpretation of the random drawing of p-values as balls being randomly placed into an ordered collection of bins, enumerated 1 up to $N + 1$. Framed in this way, we see that $BH_q$ operates by identifying the *rejection count*

$$\tilde{k} = \max\left\{ i \in [0, N]_{\mathbb{Z}} : B_{1:i}^{\mathcal{N}} = i \right\}, \tag{2}$$

which corresponds to the largest collection of consecutive bins $1, \ldots, i$ that collectively contain precisely $i$ balls, so that $BH_q$ rejects all tests with p-values lying in the first $\tilde{k}$ bins. The case $\tilde{k} = 0$ corresponds to rejecting no tests. In fact, $\tilde{k}$ is a stopping time under a filtration $\mathcal{F}$ that we define next.

**Filtration** $\mathcal{F} = \{\mathcal{F}_i\}_{i=0}^N$**:** Let $\Omega := [0, 1]^N$ be a sample space with the classical Borel $\sigma$-algebra $\mathcal{B}$ and (with slight abuse of notation) probability measure $\mathbb{P} := (\otimes_{i \in \mathcal{H}_0} U(0, 1)) \otimes (\otimes_{i \in \mathcal{H}_1} \mathbb{P}_i^1)$. We define a filtration beginning with $\mathcal{F}_N := \sigma(B_{N+1}^{\mathcal{N}}, B_{N+1}^0)$, and continuing inductively ($N$ towards 0), let $\mathcal{F}_i$ be the $\sigma$-algebra generated by $\{B_j^{\mathcal{N}}\}_{j=i+1}^{N+1}$ and $\{B_j^0\}_{j=i+1}^{N+1}$. In words, this filtration corresponds to what is cumulatively learned about the bin loads (null and total) upon examination of the bins in sequence starting with bin $N + 1$ and concluding with bin 1, assuming each observed p value comes with correct identification of whether or not $i \in \mathcal{H}_0$.

We note the fact that for $\ell > i$, it follows that $E\left[ \frac{B_{1:i}^0}{i} \middle\| \mathcal{F}_\ell \right] = E\left[ \frac{B_{1:i}^0}{i} \middle\| \frac{B_{1:\ell}^0}{\ell} \right] = \frac{B_{1:\ell}^0}{\ell}$ a.s., so that $\frac{B_{1:N}^0}{N}, \frac{B_{1:N-1}^0}{N-1}, \ldots, \frac{B_{1:1}^0}{1}$ form a martingale sequence adapted to the filtration. This fact will prove useful when combined with the optional stopping theorem to facilitate several results in this work.

Under this lens, the adversary's (algorithmic) task reduces to reshuffling p-values among the bins, and in Section 3 we demonstrate there are indeed simple, tractable ways of performing this to manipulate

BH, with potentially great effect on FDR control. The key insight is that there exist alternative stopping times that present alternative rejection counts (/regions) that can break FDR control.

# 3 Adversarial Algorithm: INCREASE-c

Throughout this section, we don the role of the adversary and study the c-Perturb problem, in which we are given $q \in (0,1)$, a realized collection $p = \{p_i\}_{i \in \mathcal{N}}$ (along with the labels of null or alternative for each $p_i$), and a budget $c \geq 1$, and our task is to produce a perturbed collection $p'$. Toward this, we focus on a procedure called INCREASE-c that despite its sub-optimality (see Appendix Section 7.2) is intuitive and simple to execute; further, as theoretical and empirical analysis in sections 4 and 5 respectively show, it has strong performance in expectation.

We begin by defining a random variable that is a stopping time adapted to $\mathcal{F}$; given an integer $c \geq 1$, let

$$\tilde{k}_{+c} := \begin{cases} \max\{i \in [c, N]_{\mathbb{Z}} : B_{1:i}^{\mathcal{N}} = i - c\} & B_{N+1}^0 \geq c \\ \tilde{k} & B_{N+1}^0 < c, \end{cases} \tag{3}$$

and we choose to write $\tilde{k}_+$ in place of $\tilde{k}_{+1}$.

**Increasing the Rejection Count** The interest in $\tilde{k}_{+c}$ is that if we moved any selection of $c$ null p-values from bin $N + 1$ into bin $\tilde{k}_{+c}$ (in fact any bin $i \leq \tilde{k}_{+c}$), then $BH_q$ would output a new, increased rejection count $\tilde{k}_{+c}$. We formally study this in Section 4. In the meantime, we comment on the increase $\tilde{k}_{+c} - \tilde{k}$, which is a difference between two stopping times

It is easy to see that $\tilde{k}_{+c} - \tilde{k} \geq c$ whenever $B_{N+1}^0 \geq c$; hence, the increase in the rejection count is at least $c$, but possibly more. We provide a stronger lower bound on this increase by utilizing the ratio between the number $B_{\tilde{k}+2:N}^0$ of nulls not rejected by $BH_q$ and the number $N - (\tilde{k} + 1)$ of bins left outside of the $BH_q$ rejection region in the case of no corruption. Computational experiments indicate comparable performance of this bound with those of simulations presented in Section 5's Table 1.

**Theorem 3.1.** *If $c \geq 1$, then*

$$\mathbb{E}\left[\tilde{k}_{+c} - \tilde{k} \| B_{N+1}^0 \geq c\right] \geq \frac{c - 1}{1 - \mathbb{E}\left[\frac{B_{\tilde{k}+2:N}^0}{N - (\tilde{k}+1)} \| B_{N+1}^0 \geq c\right]} + 1 \tag{4}$$

*for any collection of alternative hypothesis distributions $\{\mathbb{P}_i^1\}_{i \in \mathcal{H}_1}$.*

INCREASE-c runs as follows:

1. IF $B_{N+1}^0 \geq c$, then move the largest $c$ (ties broken arbitrarily) in the (N+1)-th bin to bin $\tilde{k}_{+c}$.
2. ELSE leave the p-values unperturbed.

It in fact suffices for the $c-$many p-values to be placed in any bin $i \leq \tilde{k}_{+c}$. We remark that since an oblivious adversary cannot discern null-drawn from alternative-drawn in the collection $p$, INCREASE-c as written is unimplementable in such a case. Hence, for the oblivious adversary, we modify INCREASE-c's criterion to $B_{N+1}^{\mathcal{N}} \geq c$ and have the oblivious adversary now take the $c-$many p-values uniformly at random from among the p-values in the $(N + 1)$-th bin. Intuitively, this modification for the oblivious adversary should yield nearly $c$ null p-values being moved (on average) just as in the non-oblivious case, assuming the proportion of nulls among the $B_{N+1}^{\mathcal{N}}$ - many p-values is high, as a typically large $\pi_0$ would entail.

We conclude this section with a characterization of the average increase in FDR, denoted $\Delta_c$ that INCREASE-c induces.

**Theorem 3.2.** *Given $c \geq 1$, let $p_{+c}$ denote the perturbed form of $p$ that INCREASE-c produces. Then the adversarially-adjusted FDR induced by INCREASE-c is*

$$\mathbb{E}FDP[BH_q; p_{+c}] = \mathbb{E}FDP[BH_q; p] + \Delta_c,$$

*for any collection of alternative distributions $\{\mathbb{P}_i^1\}_{i \in \mathcal{H}_1}$, where*

$$\Delta_c := \mathbb{E}\left[\frac{c}{\tilde{k}_{+c}}; B_{N+1}^0 \geq c\right]. \tag{5}$$

In Section 4 to follow, we provide analytical lower bounds for $\Delta_c$ as part of a discussion on BH's adversarial robustness. Section 5 presents computational experiments (e.g. Table 1). We remark that INCREASE-c is not optimal for all instances of c-Perturb; indeed, for $c = 1$, we present a provably optimal algorithm MOVE-1 in Appendix Section 7.2, which in contrast to INCREASE-1 sometimes induces a reduced rejection count. However, INCREASE-c remains a formidable adversarial procedure, as the results of Section 5 demonstrate on not only i.i.d. p-values but also PRDS conformal p-values.

## 4 Theoretical Analysis: Performance Guarantees and Insights into Adversarial Robustness

BH's FDR control – Lemma 1.1 – is (distributionally) robust in the sense that it holds no matter the alternative distributions $\{\mathbb{P}_i^1\}_{i \in \mathcal{H}_1}$. However, as Theorem 3.2 indicates, the degree to which this control can withstand data perturbations at test time, i.e., its adversarial robustness, very much depends on $\{\mathbb{P}_i^1\}_{i \in \mathcal{H}_1}$.

Recalling Lemma 1.1, we may assume without loss of generality that no alternative distribution $\mathbb{P}_i^1$ stochastically dominates $U(0,1)$ (equiv., $\mathbb{P}_i^1 \succcurlyeq U(0,1)$). That being said, the "degree" to which the alternative distributions $\{\mathbb{P}_i^1\}_{i \in \mathcal{H}_1}$ are (stoch.) dominated by the null distribution $U(0,1)$ (equiv., $\mathbb{P}_i^1 \preccurlyeq U(0,1)$) is critical. We briefly preview two regimes of special interest for which each of the next two subsections cover.

**High sub-uniformity:** When the alternatives are sub-uniform $\mathbb{P}_i^1 \preccurlyeq U(0,1)$ for all $i \in \mathcal{H}_1$, and highly so, such that for all $i \in \mathcal{H}_1$ it holds that $\mathbb{P}_i^1 (p_i < \epsilon) \approx 1$ for some small $\epsilon > 0$, then it follows that $\tilde{k}$ is large and $B_{1:\tilde{k}}$ should contain most alternative p-values. Consequently, in order for INCREAES-c to induce any sizeable increase to the FDR, the adversary will need to expand the BH rejection region significantly so as to introduce a commensurate number of nulls. Table 1 indicates $c$ may need to be quite large to make a dent in FDR control. This message is made more precise in Theorem 4.1.

**Low sub-uniformity:** As we will see, when the alternative p-values are barely dominated by $U(0,1)$, $\Delta_c$ can be rather large. In fact, in the special case that there is no dominance such that $\mathbb{P}_i^1 = U(0,1)$ for all $i \in \mathcal{H}_1$, a strikingly vulnerable state occurs with high probability. Indeed, in this case where nulls and alternatives are virtually indistinguishable, $BH_q$ (in fact any $\mathcal{A}$) admits an FDR of $\pi_0$ whenever any rejections are made (i.e., $\mathbb{E}\left[FDP[BH_q; p] \| \tilde{k} \geq 1\right] = \pi_0$) so that $BH_q$ accordingly compensates by making no rejections with high probability ($\mathbb{P}\left(\tilde{k} = 0\right) = 1 - q$), which follows by the distributional robust control (1) from Lemma 1.1. But those times when $\tilde{k} = 0$ is precisely when INCREASE-c's simultaneous expansion of the rejection region and injection of nulls into this region is most damaging. That this event and other similarly vulnerable events occurs with high probability is the fault of the distributional robustness. This message is made rigorous in the forthcoming Theorem 4.5.

### 4.1 Case of High Sub-Uniformity in Alternatives $\{\mathbb{P}_i^1\}_{i \in \mathcal{H}_1}$

If $\mathbb{P}_i^1 \preccurlyeq U(0,1)$ for all $i \in \mathcal{H}_1$, with $\mathbb{P}_i^1 (p_i < \epsilon) \approx 1$ for some small $\epsilon > 0$, then it is clear that the number of alternatives rejected by $BH_q$ should be nearly the maximum number $N_1$ of correct rejections possible (i.e., $B_{1:\tilde{k}}^1 \approx N_1$) with high probability, limiting any potential impact of INCREASE-c.

The following bounds are formulated to elaborate on such dynamics in this case of large separation between alternatives and nulls, for which the event $\left[B_{1:c}^1 = N_1\right]$ has probability close to 1.

**Theorem 4.1.** *If $c \geq 1$, then*

$$\Delta_c \leq \mathbb{P}\left(B_{1:c}^1 = N_1\right) \mathbb{E}\left[\frac{c}{c + N_1 + B_{1:N_1+c}^0} \| B_{N+1}^0 \geq c\right] + 1 - \mathbb{P}\left(B_{1:c}^1 = N_1\right)$$

*and*

$$\mathbb{E}\left[\tilde{k}_{+c} \| B_{N+1}^0 \geq c\right] \geq \frac{(N_1 + c) \cdot \mathbb{P}\left(B_{1:c}^1 = N_1\right)}{1 - \mathbb{E}\left[\frac{B_{1:N}^0}{N} \| B_{1:N}^0 \leq N_0 - c\right]}. \tag{6}$$

In words, for fixed $c$, as the alternative distributions concentrate more and more on $0$, it follows that $\mathbb{P}\left(B_{1:c}^1 = N_1\right) \uparrow 1$, so that the effect $\Delta_c$ of INCREASE-c on BH's FDR is dampened. And this occurs despite the fact that the increase $\tilde{k}_{+c} - \tilde{k}$ in rejection count produced by INCREASE-c consists of mostly the introduction of nulls, and tends to a magnification of $(N_1 + c)$ by at least a factor of the inverse of $1 - \mathbb{E}\left[\frac{B_{1:N}^0}{N} \| B_{1:N}^0 \le N_0 - c\right]$, which is straightforward to compute since $B_{1:N}^0 \sim Binom(N_0, q)$. We refer the reader to Section 5's Table 1 for simulations illustrating the above.

## 4.2 Case of Low Sub-Uniformity in Alternatives $\{\mathbb{P}_i^1\}_{i \in \mathcal{H}_1}$

In the study of this regime, we aim to demonstrate that the adversarial increase $\Delta_c$ can be rather large. We provide a lower bound $L_c$ on $\Delta_c$ that will be a function of parameters $q, N, N_0$, as well as the alternative distributions $\{\mathbb{P}_i^1\}_{i \in \mathcal{H}_1}$.

### 4.2.1 Lower Bounding $\Delta_c$

In view of 5, the event $[\tilde{k}_{+c} = c]$ is clearly of significance in the computation of $\Delta_c$. In words, this event describes the case that the $BH_q$ rejection region captures nothing, and yet when the adversary successfully executes INCREASE-c the rejection will now capture only nulls - generating a false detection proportion of 1. Hence, we lower bound the adjustment $\Delta_c$ of Theorem 3.2 by lower-bounding the probability of this event.

Our strategy: (1) first, characterize this probability under the special case that $\mathbb{P}_i^1 = U(0, 1)$ for all $i \in \mathcal{H}_1$; (2) second, to handle when $\mathbb{P}_i^1 \preccurlyeq U(0, 1)$ for some $i \in \mathcal{H}_1$, we translate the KL divergence of the resulting discrete, bin-assignment distributions into a bound via Pinsker's Inequality.

The key to the first step will be to recognize that when $\mathbb{P}_i^1 = U(0, 1)$ for all $i \in \mathcal{H}_1$, the vector of total loads $(B_1^{\mathcal{N}}, \ldots, B_N^{\mathcal{N}})$ is exchangeable (i.e., its law is invariant under permutations), given $B_{1:N}^{\mathcal{N}}$. Indeed, exchangeability, combined with the generalized Ballot Theorem of [31] yields the following result:

**Corollary 4.2.** *Let $n \ge 1$, $p \in [0, 1]$, and $x$ a non-negative integer such that $0 \le x \le n$. If $\tilde{B} = (\tilde{B}_1, \ldots, \tilde{B}_n) \sim Multinomial\left(x, (p, \ldots, p)\right)$, then $\mathbb{P}_{\tilde{B}}\left(\cap_{r=1}^n \left[\sum_{i=1}^r \tilde{B}_i < r\right]\right) = 1 - \frac{x}{n}$.*

Armed with Corollary 4.2, we can begin to estimate the probability laws of $\tilde{k}$ and $\tilde{k}_{+c}$.

**Corollary 4.3.** *If $\mathbb{P}_i^1 = U(0, 1)$ for all $i \in \mathcal{H}_1$,*
*then $\mathbb{P}\left(\tilde{k} = \ell\right) = \left(\frac{1-q}{1-q+\frac{(N-\ell)q}{N}}\right) \cdot \mathbb{P}\left(B_{1:\ell}^{\mathcal{N}} = \ell\right)$ for $\ell = 0, \ldots, N$, where $0^0 = 1$,*
*and $\mathbb{P}\left(B_{1:\ell}^{\mathcal{N}} = \ell\right) = \binom{N}{\ell}\left(\frac{q\ell}{N}\right)^\ell \left(1 - \frac{q\ell}{N}\right)^{N-\ell}$.*
*If $B_{N+1}^0 \ge c$, then $\mathbb{P}\left(\tilde{k}_{+c} = c \;\|\; B_{N+1}^0\right) = \left(1 - \frac{cq}{N}\right)^{N_1} \cdot \left(1 - \frac{c}{N}\right)^{N_0 - B_{N+1}^0}\left(1 - \frac{N_0 - B_{N+1}^0}{N-c} - \frac{N_1 q}{N-cq}\right)$.*

Next, we carry out the second step of our plan. Towards this, we make the following assumption.

**Assumption 4.4.** Suppose the alternative p-values are independent and identically distributed, with common distribution $\mathbb{P}^1$, i.e., $\{p_i\}_{i \in \mathcal{H}_1} \overset{iid}{\sim} \mathbb{P}^1$. We write $\delta := (1 - q) - \mathbb{P}^1(p_i \in B_{N+1})$, and $\delta_j := \mathbb{P}^1(p_i \in B_j) - \frac{q}{N}$, and we assume that $\mathbb{P}^1(p_i \in B_j) > 0$ for all $j \in [N]$, as well as $\mathbb{P}^1(p_i \in B_{\ell:N}) > 0$ for arbitrary $\ell < N$.

This assumption not only reduces the notational burden (otherwise documenting $N_1$ many alternative distributions) but it also allows the leveraging of KL divergences between the different bin assignment distributions implied by the alternative $\mathbb{P}^1$ versus the null $U(0, 1)$.

**Theorem 4.5.** *Suppose Assumption 4.4. Then*

$$\Delta_c \ge L_c(q, N, N_0, \mathbb{P}^1) := \left(1 - \frac{cq}{N} - \delta_{1:c}\right)^{N_1}\left[\left(1 - \frac{c}{N}\right)^{N_0}\left(1 - \pi_c - V_c\right)M_c + Z_c\right]$$

*where $\pi_c(q, N, N_0, \mathbb{P}^1) := \frac{N_0 + \mathbb{E}[B_{c+1:N}^1 \| B_{1:c}^1 = 0]}{N-c}$, $\quad \mathbb{E}[B_{c+1:N}^1 \| B_{1:c}^1 = 0] = N_1 \frac{(N-c)q + N\delta_{c+1:N}}{N-cq-N\delta_{1:c}}$,*
*$V_c(q, N, N_0, \mathbb{P}^1) := \sqrt{\frac{ln2}{2}\mathbb{E}[B_{c+1:N}^1 \| B_{1:c}^1 = 0] D_{KL}(\mathbb{P}^1, c)}$,*

$$M_c(q, N, N_0) := \frac{N - cq}{N - c} - \mathbb{E}\left[\left(1 - \frac{c}{N}\right)^{-B^0_{N+1}}; B^0_{N+1} \leq c - 1\right],$$

$$Z_c(q, N, N_0) := \mathbb{P}\left(B^0_{N+1} \geq c\right)\left(1 - \frac{c}{N}\right)^{N_0 - \mathbb{E}[B^0_{N+1} \| B^0_{N+1} \geq c]} \frac{\mathbb{E}[B^0_{N+1} \| B^0_{N+1} \geq c]}{N - c}$$

$$D_{KL}(\mathbb{P}^1, c) := \sum_j \frac{1}{N - c} \log\left(\frac{q + \delta - \sum_{j=1}^c (q/N + \delta_j)}{(N - c)(q/N + \delta_j)}\right)$$

*Remark* 4.6. For a more concrete application/example, consider when the p-values $\{p_i\}_{i \in \mathcal{N}}$ are derived from z-scores $\{z_i\}_{i \in \mathcal{N}}$ via $p_i = \mathbb{P}(Z > z_i)$, where $Z \sim N(0, 1)$, with null z-scores $\{z_i\}_{i \in \mathcal{H}_0}$ i.i.d. $N(0, 1)$ and alternative z-scores $\{z_i \sim N(\mu_1^i, 1)\}_{i \in \mathcal{H}_1}$ (with all $\mu_1^i \geq 0$). Indeed, under this framework, we may view the $\mu_1^i$ as the distance between $\mathbb{P}_i^1$ and $U(0, 1)$.

Towards satisfying Assumption 4.4, we can assume throughout that there exists a nonnegative parameter $\mu_1$ such that the alternative parameters $(\mu_1^i, \sigma^i) = (\mu_1, 1)$ for all $i \in \mathcal{H}_1$. Indeed, if every member of the collection $\{\mu_1^i\}_{i \in \mathcal{H}_1}$ is in reality only close to zero, then the forthcoming estimates (lower bounds) would provide close, conservative estimates when setting $\mu_1 = \max\{\mu_1^i : i \in \mathcal{H}_1\}$.

When $\mu_1 > 0$, it follows that $\delta(\mu_1) := (1 - q) - \Phi\left(\frac{\Phi^{-1}(1-q) - \mu_1}{\sigma_1}\right) > 0$. As well, $\delta_j(\mu_1) :=$ $\Phi\left(\frac{\Phi^{-1}(1 - \frac{(j-1)q}{N}) - \mu_1}{\sigma_1}\right) - \Phi\left(\frac{\Phi^{-1}(1 - j\frac{q}{N}) - \mu_1}{\sigma_1}\right) - \frac{q}{N}$. Intuitively, BH is adversarially robust ($\Delta_c$ small) when the alternative distributions are "far" ($\mu_1 >> 0$) from the theoretical null, and the opposite holds when they are "close" ($\mu_1 = 0$) - see Figures 2 and 3.

# 5 Simulations and Data Experiments

In this section, we provide computations (performed in R and Python on a Macbook Air-M2 chip, 8GB memory, with no experiment time exceeding 5 minutes) to demonstrate the performance of the adversarial algorithm INCREASE-c. We demonstrate its performance through simulation on synthetic data to make comparisons to the theoretical estimates provided in Section 4. We then demonstrate its performance on a real-data experiment in outlier detection.

## 5.1 INCREASE-c Simulations on Synthetic Data

### 5.1.1 INCREASE-c Simulations on i.i.d. p-values

Following the framework from Remark 4.6, we simulated $10^4$ replications of the following experiment: (1) $N = 1000$ p-values are generated, with $\{p_i\}_{i \in \mathcal{N}} \overset{iid}{\sim} U(0, 1)$, and each $p_i$ among $\{p_i\}_{i \in \mathcal{N}_1}$ generated via $p_i = 1 - \Phi(\frac{z_i - \mu_1}{1})$, with $\{z_i\}_{i \in \mathcal{N}_0} \overset{iid}{\sim} N(\mu_1, 1)$; (2) $FDP[BH_q; p]$ and $FDP[BH_q; p_{+c}]$ are calculated. In Figure 1 each of the $10^4$- many $(FDP[BH_q; p], FDP[BH_q; p_{+c}])$ pairs are plotted. As can be seen, the vast majority of the pairs satisfy $FDP[BH_q; p_{+c}] > FDP[BH_q; p]$, and, further, all pairs lie above the horizontal line situated at the level of the BH control level $\pi_0 \cdot q = .09$, i.e, $FDP[BH_q; p_{+c}] > \pi_0 \cdot q$.

In Table 1, we present the effectiveness of INCREASE-c over ranges of corruption budget $c$ and $\mu_1$ from small to large. As can be seen, when $\mu_1 = 0$, any amount of corruption budget $c$ yields large post-corruption FDR $\mathbb{E}_z FDP[BH_q; z_{+c}]$. When $\mu_1 > 0$ grows, however, the budget $c$ must correspondingly grow in order for there to be nontrivial post-corruption FDR. Finally, we note that for any fixed $c$, the increase in rejection count $\tilde{k}_{+c} - \tilde{k}$ is on the average larger when $\mu_1$ is larger. For experiments on non- i.i.d., PRDS p-values, we refer the interested reader to Section 5.2 or Appendix Section 7.4.

### 5.1.2 INCREASE-1 Simulations versus Theoretical Bounds Under Small $\mu_1$

In Figures 2 and 3 we illustrate how well the insights discussed in Section 4.2 capture the sensitivity of BH to adversarial perturbations when $\mu_1$ is near 0. Specifically, for each $q$ in the grid $\{.01, .02, \ldots, .99\}$, we computed the difference $FDP[BH_q; p_+] - FDP[BH_q; p]$ across $10^3$ replications of the setup as in Section 5.1.1, with the average of this difference being an estimate of $\Delta_1$. The plot of these $\Delta_1$ estimates with respect to $q$ is then compared with the plot of our lower bound $L_1$ as a function of $q$.

Figures 2 and 3 illustrate that the stricter the control, i.e., the smaller $q$ is, the more effective INCREASE-c becomes. In fact, the bound indicates high vulnerability for the typical use case

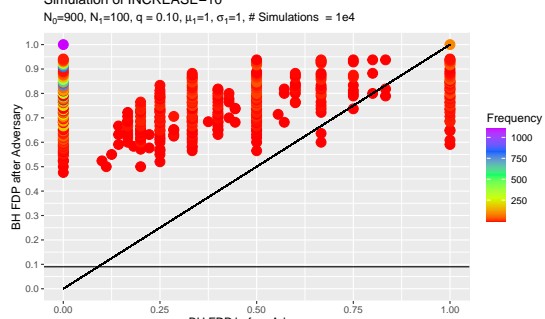

Simulation of INCREASE−10
$N_0=900$, $N_1=100$, q = 0.10, $\mu_1=1$, $\sigma_1=1$, # Simulations = 1e4

| $c$ | $\mu_1$ | 0 | 1 | 2 |
|---|---|---|---|---|
| 1 | | .99 (1.11) | .75 (1.39) | .14 (1.9) |
| 2 | | .99 (2.22) | .77 (2.75) | .18 (3.7) |
| 5 | | .99 (5.6) | .81 (6.67) | .26 (9.0) |
| 10 | | .99 (11.22) | .84 (13.76) | .36 (17.0) |
| 100 | | .99 (111.30) | .91 (121.48) | .72 (136.2) |
| 200 | | .99 (222.41) | .93 (237.78) | .81 (256.0) |
| 500 | | .99 (555.59) | .95 (580.75) | .89 (601.5) |

Figure 1: $10^4$ simulations of FDP by $BH_q$ before and after INCREASE-10 is executed on the p-values. $N = 10^3$, $N_0 = 900$, and $q = 0.10$.

Table 1: Sample Average ($10^4$-batch) estimates of $\mathbb{E}_z FDP[BH_q; z_{+c}]$ and $\mathbb{E}\left[\tilde{k}_{+c} - \tilde{k}\right]$ (in parentheses) when all $\{\mu_1^i\}_{i \in \mathcal{H}_1}$ commonly equal some $\mu_1 \in \{0, 1, 2\}$ and $N = 10^3$, $q = 0.10$, $\pi_0 = 0.90$, and all $\sigma^i = 1$.

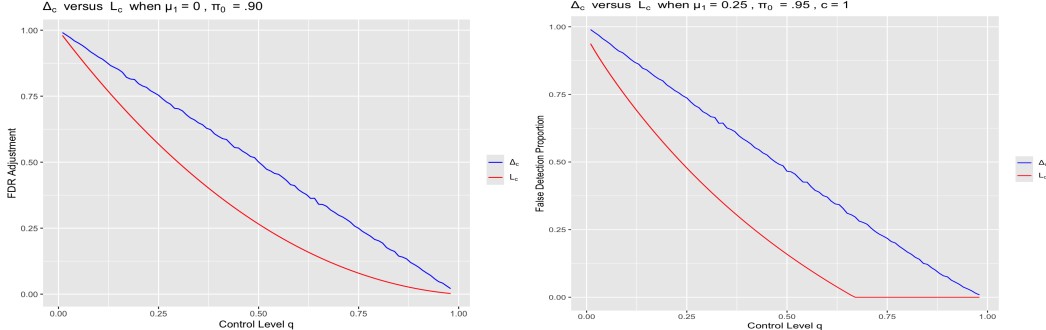

Figure 2: Comparing the FDR increase $\Delta_1$ of INCREASE-1 with the lower bound $L_1$ of Theorem 4.5 as functions of $q$ when $\mu_1 = 0$, $N = 1000$

Figure 3: Comparing the FDR increase $\Delta_1$ of INCREASE-1 with the lower bound $L_1$ of Theorem 4.5 as functions of $q$ when $\mu_1 = .25$, $N = 1000$

of $q \in (0, 0.10)$. On the other hand, as $q$ increases, INCREASE-c's effect weakens; however, for increased $q$, the decision maker is accepting a higher FDR already.

## 5.2   Real-Data Experiment: Credit Card Fraud Detection

The `Credit Card`[2] dataset $D := \{(X_i, Y_i)\}_{i=1}^n \subseteq \mathbb{R}^{30} \times \{0, 1\}$ contains $n = 284,807$ credit card transactions in September 2013 by European cardholders over the course of two days - 492 of which were frauds. Each $X_i \in \mathbb{R}^{30}$ consists of numerical input variables that are the result of a PCA transformation. The "Class" label $Y_i$ takes value 1 in case of fraud and 0 otherwise (/genuine), yielding the partition $[n] = n_0 \uplus n_1$, with $|n_0| = 284,315$ and $|n_1| = 492$.

### Fraud Detection Experiment

Given a set of unlabeled transactions $\{X_i\}_{i \in S}$, where $S \subseteq [n]$, the *fraud detection* task is to identify which members of $S$ belong to $n_1$, i.e., are fraudulent. We experiment with the BH-based, outlier detection method of [5] on this fraud detection task - specifically, we consider the false detection proportion of this method in the absence and presence of an adversary. The details of the experimental setup will now be discussed.

---

[2]https://www.kaggle.com/datasets/mlg-ulb/creditcardfraud

**Training**

We begin by training an unsupervised decision-tree-based algorithm on a training set. From the set $n_0$ of genuine transactions, we uniformly at random select a subset $n_{train} \subseteq n_0$ of size $141,758$ to form a training set $D^{train} := \{X_i\}_{i \in n_{train}}$ upon which we train an isolation forest [25] $\hat{s} : \mathbb{R}^{30} \to \mathbb{R}_+$ using the R library `isotree`[3], where, in principle, $\hat{s}(X_i)$ returns an *isolation depth* that is smaller if $Y_i = 1$ (i.e. is an outlier) and larger if $Y_i = 0$ (i.e. is an inlier)

**Calibration and (Adversarially-Perturbed) Testing**

Then for each of $10^2$ simulations, we uniformly at random selected a subset $\tilde{n}_{cal} \subseteq n_0 \setminus n_{train}$ of size $141,657$ to form a calibration set $\tilde{D}^{cal} := \{X_i\}_{i \in \tilde{n}_{cal}}$ of strictly genuine transactions. As well, we uniformly at random selected a subset $\tilde{n}_{test,1} \subseteq n_1$ of 100 fraudulent transactions to append to the 900 remaining genuine transactions comprising $\tilde{n}_{test,0} := n_0 \setminus n_{train} \setminus \tilde{n}_{cal}$ to form a test set $\tilde{D}^{test} := \{X_i\}_{i \in \tilde{n}_{test}}$, where $\tilde{n}_{test} := \tilde{n}_{test,0} \cup \tilde{n}_{test,1}$. Finally, we transformed $X_i \mapsto \tilde{p}_i \in (0,1)$ for each $i \in \tilde{n}_{test}$ via $\tilde{p}_i = \frac{1 + |\{j \in \tilde{n}_{cal} : \hat{s}(X_j) \leq \hat{s}(X_i)\}|}{|\tilde{n}_{cal}|}$. The resulting collection of conformal p-values $\tilde{p} := (\tilde{p}_i)_{i \in \tilde{n}_{test}}$ is PRDS, as proven in [5], and hence the FDR control of Lemma 1.1 holds (see [7]). In contrast, upon executing INCREASE-c to generate a corresponding adversarially-perturbed collection $\tilde{p}_{+c} := (\tilde{p}_{+c,i})_{i \in \tilde{n}_{test}}$, we obtain a collection for which BH's FDR control no longer holds.

**Experimental Results**

We executed $BH_{0.1}$, on both $\tilde{p}$ and $\tilde{p}_{+c}$, with an execution providing the decision for each $i \in \tilde{n}_{test}$ whether to report it as genuine (null) or fraudulent (alternative). We report the average (over the $10^2$ simulations) false detection proportion (FDP) produced by $BH_{0.1}$, i.e., both $E[FDP[BH_{0.1}; \tilde{p}]]$ and $E[FDP[BH_{0.1}; \tilde{p}_{+c}]]$ (for $c = 1, 5, 10$). As well, we report the average number of alleged frauds $\mathbb{E}[\tilde{k}]$ and $\mathbb{E}[\tilde{k}_{+c}]$. As Table 2 indicates, although the method of [5] can ordinarily control the

| $c$ | $E[FDP[BH_{0.1}; \tilde{p}]]$ | $E[FDP[BH_{0.1}; \tilde{p}_{+c}]]$ | $E[\tilde{k}]$ | $E[\tilde{k}_{+c}]$ |
|---|---|---|---|---|
| 1 | .09 | .10 | 60.21 | 60.21 |
| 5 | .08 | .17 | 48.69 | 57.12 |
| 10 | .09 | .23 | 56.39 | 72.85 |
| 20 | 0.09 | 0.31 | 58.12 | 89.06 |

Table 2: Credit Card Fraud Detection Experiment

FDR below the explicit $0.10$ level, INCREASE-c has the ability to push the FDR above this control level.

## 6 Conclusions

This is the first work to consider adversarial corruption of the popular Benjamini Hochberg multiple testing procedure to break its FDR control. While BH may exhibit robustness when the alternative distributions are "far" from the null, it exhibits great sensitivity in practical cases when the alternatives are "closer" to the null. In such cases, with the modification of few p-values (as few as one), the attacker can increase the expected FDR well past the guarantee stipulated by the BH procedure. This study suggests some caution may be necessary when using BH, especially in safety-security settings. Numerical experiments support the analytical results. Finally, BH is but one member of the family of step-up multiple testing procedures, which generally entail rejection regions decided via a stopping time, which since our paper shows can be manipulated in the case of BH, it means other step-up procedures can be similarly prone.

---

[3]https://cran.r-project.org/web/packages/isotree/vignettes/An_Introduction_to_Isolation_Forests.html

## Acknowledgments and Disclosure of Funding

We wish to thank Wang Chi Cheung for a stimulating conversation. This work was supported by the Air Force Office of Scientific Research (Mathematical Optimization Program) under the grant: "Optimal Decision Making under Tight Performance Requirements in Adversarial and Uncertain Environments: Insight from Rockafellian Functions".

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

# 7 Appendix / supplemental material

## 7.1 Adversarial Algorithm: INCREASE-c

**Theorem 3.1.** *If $c \geq 1$, then*

$$\mathbb{E}\left[\tilde{k}_{+c} - \tilde{k} \| B_{N+1}^0 \geq c\right] \geq \frac{c-1}{1 - \mathbb{E}\left[\frac{B_{\tilde{k}+2:N}^0}{N - (\tilde{k}+1)} \;\middle\|\; B_{N+1}^0 \geq c\right]} + 1 \tag{4}$$

*for any collection of alternative hypothesis distributions $\{\mathbb{P}_i^1\}_{i \in \mathcal{H}_1}$.*

*Proof.* Since the statement in the case of $c = 1$ is trivially true, we henceforth assume that $c > 1$. Recall $\tilde{k}_{+c} = \max\{i \in [c, N]_{\mathbb{Z}} : B_{1:i} = i - c\}$. Let us define $\tilde{k}_{+c}^0 := \max\{i \in [c, N]_{\mathbb{Z}} : B_{1:i}^0 = i - (B_{1:\tilde{k}}^1 + c)\}$. Then we begin by establishing that

$$\tilde{k}_{+c} \geq \tilde{k}_{+c}^0 > \tilde{k} + 1.$$

To see why this holds, we note that $B_{1:\tilde{k}+1}^0 = \tilde{k} - B_{1:\tilde{k}+1}^1 = \tilde{k} + 1 - (B_{1:\tilde{k}+1}^1 + 1) > \tilde{k} + 1 - (B_{1:\tilde{k}+1}^1 + c)$, and this implies $\tilde{k} + 1 < \tilde{k}_{+c}^0$. Further, $B_{1:\tilde{k}_{+c}^0} = B_{1:\tilde{k}_{+c}^0}^0 + B_{1:\tilde{k}_{+c}^0}^1 = \tilde{k}_{+c}^0 - (B_{1:\tilde{k}}^1 + c) + B_{1:\tilde{k}_{+c}^0}^1 = \tilde{k}_{+c}^0 - c + (B_{1:\tilde{k}_{+c}^0}^1 - B_{1:\tilde{k}}^1) \geq \tilde{k}_{+c}^0 - c$, which implies $\tilde{k}_{+c} \geq \tilde{k}_{+c}^0$.

Next, we justify the relation

$$\mathbb{E}\left[\frac{B_{\tilde{k}+2:N}^0}{N - (\tilde{k}+1)} \;\middle\|\; \tilde{k}, B_{1:\tilde{k}}^1, [B_{N+1}^0 \geq c]\right] = \mathbb{E}\left[\frac{B_{\tilde{k}+2:\tilde{k}_{+c}^0}^0}{\tilde{k}_{+c}^0 - (\tilde{k}+1)} \;\middle\|\; \tilde{k}, B_{1:\tilde{k}}^1, [B_{N+1}^0 \geq c]\right].$$

It suffices to establish it for any fixed, joint realization of $\tilde{k}, B_{1:\tilde{k}}^1$ that occurs consistent with the event $[B_{N+1}^0 \geq c]$ with positive probability. With slight abuse of notation, we will continue to use $\tilde{k}, B_{1:\tilde{k}}^1$ for such a fixed realization, and write $\bar{\mathbb{E}}[\cdot]$ for $\mathbb{E}\left[\cdot \| \tilde{k}, B_{1:\tilde{k}}^1, [B_{N+1}^0 \geq c]\right]$. The plan is to apply the Optional Stopping Theorem on a martingale sequence. To do so, let us form a (backwards-running) filtration: let $\mathcal{F}_N$ be the sigma-algebra generated by the event $[B_{N+1}^0 \geq c]$ as well as the random variable $B_{N+1}^0$, and let $\mathcal{F}_i$ be the sigma-algebra generated by the event $[B_{N+1}^0 \geq c]$ as well as the random variables $\{B_j^0\}_{j=i+1}^{N+1}$. Then for any integers $k, \ell \in \{N, N-1, \dots 1\}$ such that $\ell > k$,

$$\bar{\mathbb{E}}\left[B_{\tilde{k}+2:[k \vee (\tilde{k}+2)]}^0 \;\middle\|\; \mathcal{F}_\ell\right] = \bar{\mathbb{E}}\left[B_{\tilde{k}+2:[k \vee (\tilde{k}+2)]}^0 \;\middle\|\; B_{\tilde{k}+2:[\ell \vee (\tilde{k}+2)]}^0\right] = \frac{[k - (\tilde{k}+1)] \vee 1}{[\ell - (\tilde{k}+1)] \vee 1} B_{\tilde{k}+2:[\ell \vee (\tilde{k}+2)]}^0,$$

where $B_{\tilde{k}+2:[\ell \vee (\tilde{k}+2)]}^0 \in \mathcal{F}_\ell$ since $B_{\tilde{k}+2:[\ell \vee (\tilde{k}+2)]}^0 = N_0 - B_{N+1}^0 - (\tilde{k} - B_{1:\tilde{k}}^1) - B_{(\ell+1) \vee (\tilde{k}+3):N}^0$ is a measurable function of the random variables in $\mathcal{F}_\ell$. In other words, the collection $\{\frac{B_{\tilde{k}+2:[k \vee (\tilde{k}+2)]}^0}{[k - (\tilde{k}+1)] \vee 1}\}$ forms a martingale under the filtration. As well, $\tilde{k}_{+c}^0$ is a stopping time. Hence, by the Optional Stopping Theorem,

$$\mathbb{E}\left[\frac{B_{\tilde{k}+2:N}^0}{N - (\tilde{k}+1)} \;\middle\|\; \tilde{k}, B_{1:\tilde{k}}^1, [B_{N+1}^0 \geq c]\right] = \bar{\mathbb{E}}\left[\frac{B_{\tilde{k}+2:N}^0}{N - (\tilde{k}+1)} \;\middle\|\; [B_{N+1}^0 \geq c]\right]$$

$$= \bar{\mathbb{E}}\left[\frac{B_{\tilde{k}+2:\tilde{k}_{+c}^0}^0}{\tilde{k}_{+c}^0 - (\tilde{k}+1)} \;\middle\|\; [B_{N+1}^0 \geq c]\right] = \mathbb{E}\left[\frac{B_{\tilde{k}+2:\tilde{k}_{+c}^0}^0}{\tilde{k}_{+c}^0 - (\tilde{k}+1)} \;\middle\|\; \tilde{k}, B_{1:\tilde{k}}^1, [B_{N+1}^0 \geq c]\right].$$

We note that the martingale property could be established even without the event $[B_{N+1}^0 \geq c]$ in the filtration but for the sake of forthcoming computations we have included it.

Next we observe that

$$B_{\tilde{k}+2:\tilde{k}_{+c}^0}^0 + \tilde{k} - B_{1:\tilde{k}}^1 = B_{\tilde{k}+2:\tilde{k}_{+c}^0}^0 + B_{1:\tilde{k}}^0 = B_{1:\tilde{k}_{+c}^0}^0 = \tilde{k}_{+c}^0 - (B_{1:\tilde{k}}^1 + c)$$

$$\implies B_{\tilde{k}+2:\tilde{k}_{+c}^0}^0 = \tilde{k}_{+c}^0 - (\tilde{k} + c),$$

we derive

$$\frac{B^0_{\tilde{k}+2:\tilde{k}^0_{+c}}}{\tilde{k}^0_{+c} - (\tilde{k}+1)} = \frac{\tilde{k}^0_{+c} - (\tilde{k}+c)}{\tilde{k}^0_{+c} - (\tilde{k}+1)} = \frac{\tilde{k}^0_{+c} - (\tilde{k}+1) - c + 1}{\tilde{k}^0_{+c} - (\tilde{k}+1)} = 1 - \frac{c-1}{\tilde{k}^0_{+c} - (\tilde{k}+1)}$$

$$\implies 0 < \mathbb{E}\left[\frac{B^0_{\tilde{k}+2:N}}{N - (\tilde{k}+1)} \;\|\; \tilde{k}, B^1_{1:\tilde{k}}, [B^0_{N+1} \geq c]\right] = 1 - (c-1)\mathbb{E}\left[\frac{1}{\tilde{k}^0_{+c} - (\tilde{k}+1)} \;\|\; \tilde{k}, B^1_{1:\tilde{k}}, [B^0_{N+1} \geq c]\right] < 1$$

$$(*)$$

$$\implies \frac{1}{\mathbb{E}\left[\tilde{k}^0_{+c} - (\tilde{k}+1) \;\|\; \tilde{k}, B^1_{1:\tilde{k}}, [B^0_{N+1} \geq c]\right]} \leq \mathbb{E}\left[\frac{1}{\tilde{k}^0_{+c} - (\tilde{k}+1)} \;\|\; \tilde{k}, B^1_{1:\tilde{k}}, [B^0_{N+1} \geq c]\right]$$

$$= \frac{1 - \mathbb{E}\left[\frac{B^0_{\tilde{k}+2:N}}{N-(\tilde{k}+1)} \;\|\; \tilde{k}, B^1_{1:\tilde{k}}, [B^0_{N+1} \geq c]\right]}{c-1}.$$

Hence,

$$\frac{c-1}{1 - \mathbb{E}\left[\frac{B^0_{\tilde{k}+2:N}}{N-(\tilde{k}+1)} \;\|\; \tilde{k}, B^1_{1:\tilde{k}}, [B^0_{N+1} \geq c]\right]} + 1 \leq \mathbb{E}\left[\tilde{k}^0_{+c} - \tilde{k} \;\|\; \tilde{k}, B^1_{1:\tilde{k}}, [B^0_{N+1} \geq c]\right] \leq \mathbb{E}\left[\tilde{k}_{+c} - \tilde{k} \;\|\; \tilde{k}, B^1_{1:\tilde{k}}, [B^0_{N+1} \geq c]\right].$$

It follows that

$$\frac{c-1}{1 - \mathbb{E}\left[\frac{B^0_{\tilde{k}+2:N}}{N-(\tilde{k}+1)} \;\|\; B^0_{N+1} \geq c\right]} + 1 = \frac{c-1}{\mathbb{E}\left[1 - \mathbb{E}\left[\frac{B^0_{\tilde{k}+2:N}}{N-(\tilde{k}+1)} \;\|\; \tilde{k}, B^1_{1:\tilde{k}}, [B^0_{N+1} \geq c]\right] \;\|\; B^0_{N+1} \geq c\right]} + 1$$

$$\leq \mathbb{E}\left[\frac{c-1}{1 - \mathbb{E}\left[\frac{B^0_{\tilde{k}+2:N}}{N-(\tilde{k}+1)} \;\|\; \tilde{k}, B^1_{1:\tilde{k}}, [B^0_{N+1} \geq c]\right]} \;\|\; B^0_{N+1} \geq c\right] + 1$$

$$\leq \mathbb{E}\left[\tilde{k}_{+c} - \tilde{k} \;\|\; B^0_{N+1} \geq c\right],$$

where we have used (conditional) Jensen's Inequality - valid because $c > 1$ and (*) ensures $1 - \mathbb{E}\left[\frac{B^0_{\tilde{k}+2:N}}{N-(\tilde{k}+1)} \;\|\; \tilde{k}, B^1_{1:\tilde{k}}, [B^0_{N+1} \geq c]\right] > 0$.

$\square$

**Theorem 3.2.** *Given $c \geq 1$, let $p_{+c}$ denote the perturbed form of $p$ that INCREASE-c produces. Then the adversarially-adjusted FDR induced by INCREASE-c is*

$$\mathbb{E}FDP[BH_q; p_{+c}] = \mathbb{E}FDP[BH_q; p] + \Delta_c,$$

*for any collection of alternative distributions $\{\mathbb{P}^1_i\}_{i \in \mathcal{H}_1}$, where*

$$\Delta_c := \mathbb{E}\left[\frac{c}{\tilde{k}_{+c}}; B^0_{N+1} \geq c\right]. \tag{5}$$

*Proof.* We derive

$$\mathbb{E}FDP[BH_q; p_+] = \mathbb{E}\left[\frac{B^0_{1:\tilde{k}_{+c}} + c}{\tilde{k}_{+c}}; B^0_{N+1} \geq c\right] + \mathbb{E}\left[\frac{B^0_{1:\tilde{k}\vee 1}}{\tilde{k} \vee 1}; B^0_{N+1} < c\right]$$

$$= \mathbb{E}\left[\frac{B^0_{1:\tilde{k}_{+c}}}{\tilde{k}_{+c}}; B^0_{N+1} \geq c\right] + \mathbb{E}\left[\frac{c}{\tilde{k}_{+c}}; B^0_{N+1} \geq c\right] + \mathbb{E}\left[\frac{B^0_{1:\tilde{k}\vee 1}}{\tilde{k} \vee 1}; B^0_{N+1} < c\right]$$

$$= \mathbb{P}\left(B^0_{N+1} \geq c\right)\mathbb{E}\left[\frac{B^0_{1:N}}{N} \middle\| B^0_{N+1} \geq c\right] + \mathbb{E}\left[\frac{c}{\tilde{k}_{+c}}; B^0_{N+1} \geq c\right] + \mathbb{P}\left(B^0_{N+1} < c\right)\mathbb{E}\left[\frac{B^0_{1:N}}{N} \middle\| B^0_{N+1} < c\right]$$

$$= \mathbb{E}\left[\frac{B^0_{1:N}}{N}\right] + \mathbb{E}\left[\frac{c}{\tilde{k}_{+c}}; B^0_{N+1} \geq c\right],$$

as desired.

$\square$

## 7.2 MOVE-1

MOVE-1 is an efficient, optimal algorithm for an (omniscient) adversary with budget $c = 1$. We begin with several small insights en route to describing MOVE-1 in full.

**Perturbing the Rejection Count**

If $p' = p$, i.e., the sample statistics are left unperturbed, we obtain a false detection proportion of $FDP[BH_q; p] = \frac{B_{1:\tilde{k}}^0}{\tilde{k}}$. If it is possible to improve on this, it can be easily shown that in the case of $c = 1$ we must induce an altered rejection count

$$\tilde{k}' := \max\left\{ i \in [0, N]_{\mathbb{Z}} : |\{p_i'\}_{i \in \mathcal{N}} \cap B_{1:i}| = i \right\}, \tag{7}$$

henceforth referred to as the *(adversarially) adjusted rejection count* resulting from the adversary's choice of $p'$. As it turns out, for the special case of $c = 1$, $\tilde{k}' \neq \tilde{k}$ is necessary if a larger FDP is to be obtained, as the following lemma indicates.

**Lemma 7.1.** *If* $\|p - p'\|_0 \leq 1$ *and* $FDP[BH_q; p'] \neq FDP[BH_q; p]$, *then* $\tilde{k}' \neq \tilde{k}$.

*Proof.* We suppose the contrary for the sake of a contradiction, i.e., $\tilde{k}' = \tilde{k}$. It follows that

$$\left|\{j \in \mathcal{N} : 0 \leq p_j' < \tilde{k}\frac{q}{N}\}\right| = \tilde{k}' = \tilde{k} = \left|\{j \in \mathcal{N} : 0 \leq p_j < \tilde{k}\frac{q}{N}\}\right|,$$

and then by $FDP[BH_q; p';] \neq FDP[BH_q; p]$, it follows that

$$\left|\{j \in \mathcal{H}_0 : 0 \leq p_j' < \tilde{k}\frac{q}{N}\}\right| \neq \left|\{j \in \mathcal{H}_0 : 0 \leq p_j < \tilde{k}\frac{q}{N}\}\right|,$$

Since $\|p - p'\|_0 \leq 1$, these two conclusions are at odds, presenting a contradiction. $\square$

Considering Lemma 7.1, the adversary's $p' \in [0, 1]^N$ decision simplifies to deciding on an adjusted rejection count $\tilde{k}'$ from among a constrained set of integers consistent with the constraint $\|p' - p\|_0 \leq 1$. An optimal $\tilde{k}'$ can indeed be larger or smaller than, or even equal to $\tilde{k}$. Hence, towards understanding this new search space, it suffices to characterize the set of feasible $\tilde{k}'$ that are larger than $\tilde{k}$, and smaller.

**Increasing the Rejection Count (c = 1)**

Towards understanding how to increase the rejection count, we first highlight the following fact that follows immediately from (2).

**Lemma 7.2.** *If* $i \in \{\tilde{k} + 1, \ldots, N\}$, *then* $B_{1:i}^{\mathcal{N}} \leq i - 1$. *In particular,* $B_{1:\tilde{k}+1}^{\mathcal{N}} = i - 1$.

*Proof.* Suppose there exists $i \in \{\tilde{k} + 1, \ldots, N\}$ such that $B_{1:i}^{\mathcal{N}} > i - 1$. If $B_{1:i}^{\mathcal{N}} = i$, then $i > \tilde{k}$ presents a contradiction of (2). However, proceeding with $B_{1:i}^{\mathcal{N}} \geq i + 1$, we see that there necessarily exists $j \in \{i + 1, \ldots, N\}$ for which $B_{1:j}^{\mathcal{N}} = j$, for if this were not the case, then $B_{1:j}^{\mathcal{N}} \geq j + 1$ for all $j \in \{i + 1, \ldots, N\}$, meaning $B_{1:N}^{\mathcal{N}} \geq N + 1$, yet another contradiction since there are only $N$ p-values. $\square$

Consequently, for $\tilde{k}' > \tilde{k}$, we require a perturbed collection $p'$ for which bin counts increase. This necessitates a *decrease* of a p-value - see Lemma 7.5 for details. Hence, we define

$$\mathcal{L} := \{i \in \{\tilde{k} + 1, \ldots, N\} : B_{1:i}^{\mathcal{N}} = i - 1\} \tag{8}$$

and note that these are precisely the positions $i$ for which the addition of one (and only one) p-value into $B_{1:i}$ through the *decrease* of a p-value brings $i$ into candidacy for the new rejection count - see equation (2).

**Proposition 7.3.** $\mathcal{L} = \{\tilde{k}' : \tilde{k}' > \tilde{k}, \|p - p'|_0 \leq 1\}$

*Proof.* To prove the first statement, we recall that by Lemma 7.2, if $i > \tilde{k}$, then $B_{1:i}^{\mathcal{N}} \leq i - 1$. So, if $i \geq \tilde{k} + 1$ with $B_{1:i}^{\mathcal{N}} < i - 1$, then no decrease of a single p-value could increase $B_{1:i}^{\mathcal{N}}$ to $i$ (Lemma 7.5), precluding the possibility of $\tilde{k}' = i$. In other words, no $i \notin \mathcal{L}$ is achievable for the new rejection count $\tilde{k}'$.

On the other hand, say we enumerate $\mathcal{L}$ with

$$i_L > i_{L-1} > \ldots > i_1 = \tilde{k} + 1.$$

For the case of $i_L$, $\tilde{k}' = i_L$ is achieved if and only if a p-value is decreased from any bin $B_j$ with $j > i_\ell$ to a bin $B_{j-s}$ where $i_L \geq j - s \geq 1$. For the case of $i_\ell$, $\tilde{k}' = i_\ell$ is achieved if and only if a p-value is decreased from any bin $B_j$ with $i_{\ell+1} \geq j > i_\ell$ to a bin $B_{j-s}$ where $i_\ell \geq j - s \geq 1$. Finally, all these movements of p-value just described are always possible for any given $p = \{p_i\}_{i \in \mathcal{N}}$. $\qquad \square$

**Decreasing the Rejection Count (c = 1)**

Analogously, it follows that the increase of a p-value is necessary for the *decrease* of the rejection count - see Lemma 7.6, and we define

$$\mathcal{R} := \left\{ i \in \{i^* + 1, \ldots, \tilde{k} - 1\} : B_{1:i}^{\mathcal{N}} = i \right\} \cup \{i^*\}, \tag{9}$$

where $i^* := 0 \vee \max\left\{ i \in \{1, \ldots, \tilde{k} - 1\} : B_{1:i}^{\mathcal{N}} = i + 1 \right\}$. We note that $\mathcal{R}$ is precisely the set of all the achievable new (decreased) rejection counts $\tilde{k}' < \tilde{k}$ we could induce with the perturbation of a single p-value.

**Proposition 7.4.** $\mathcal{R} = \{\tilde{k}' : \tilde{k} > \tilde{k}', \|p - p'\|_0 \leq 1\}$

*Proof.* We begin by proving the first statement that $\tilde{k}' < \tilde{k}$ implies $\tilde{k}' \in \mathcal{R}$. To do so, we proceed in two steps. First we show that $\tilde{k} \geq i^*$, so that $i^* \leq \tilde{k}' \leq \tilde{k} - 1$. Then we show that if $\tilde{k}' = i$ for some $i \in \{i^* + 1, \ldots, \tilde{k} - 1\}$ in which $B_{1:i}^{\mathcal{N}} \neq i$, we arrive at a contradiction.

To see that $\tilde{k}' \geq i^*$, if $i^* > \tilde{k}'$, then $B_{1:i^*}^{\mathcal{N}} = i^* + 1$ becomes $B_{1:i^*}^{\mathcal{N}} \leq i^* - 1$ by Lemma 7.2; in other words, the change in magnitude of $B_{1:i^*}^{\mathcal{N}}$ is at least 2, which contradicts Lemma 7.6. Next, if $i \in \{i^* + 1, \ldots, \tilde{k} - 1\}$, then $B_{1:i}^{\mathcal{N}} \leq i$ by the definition of $i^*$. This means if $B_{1:i}^{\mathcal{N}} \neq i$, then $B_{1:i}^{\mathcal{N}} < i$, so that Lemma 7.6 indicates $\tilde{k}'$ could not be $i$.

As for the second statement, let $\mathcal{R}$ be enumerated

$$i_R > i_{R-1} > \ldots > i_1 = i^*.$$

For the case of $i_R$, $\tilde{k}' = i_R$ is achieved if and only if a p-value is moved from bin $B_j$ with $\tilde{k} \geq j > i_R$ to a bin $B_{j+s} > \tilde{k}$. For $R - 1 \geq r \geq 2$, by Lemma 7.6 it holds that $\tilde{k}' = i_r$ is achieved if and only if a p-value is moved from a bin $B_j$ with $i_{r+1} \geq j > i_r$ to a bin $B_{j+s}$ with $j + s > \tilde{k}$. Finally, for the case of $i_1$, $\tilde{k}' = i_1 = i^*$ is achieved if and only if a p-value is moved from $B_j$ with $i^* \geq j$ to a $B_{j+s}$ with $j + s > \tilde{k}$. Finally, all these movements of p-values just described are always possible for any given $p = \{p_i\}_{i \in \mathcal{N}}$. $\qquad \square$

It follows that $\tilde{k}' \neq \tilde{k}$ if and only if $\tilde{k}' \in \mathcal{L} \cup \mathcal{R}$, so an efficient, optimal search procedure becomes straightforward. Informally, we iterate over the bins in reverse order, beginning with $N + 1$ and terminating with $i^*$. At iteration (/bin number) $i$, if $i \in \mathcal{L} \cup \mathcal{R}$, then the trivial subproblem $\max_{p':\|p-p'\|_0 \leq 1, \tilde{k}'=i} FDP[BH_q; p']$ is solved; otherwise, nothing is done. Upon termination, the best FDP encountered is the answer. This is summarized in Theorem 7.7

**Lemma 7.5** (Decreasing a p-value). *Let $p'$ be such that $\|p - p'\|_0 \leq 1$. If $p'$ is the decrease of a single p-value in $p$ from $B_j$ to $B_{j-s}$, where $j \in \{2, \ldots, N + 1\}$ and $1 \leq s \leq j - 1$, then*

- $i \in \{1, \ldots, j - s - 1\} \implies B_{1:i}^{\mathcal{N}}$ *remains constant*

- $i \in \{j - s, \ldots, j - 1\} \implies B_{1:i}^{\mathcal{N}}$ *increases by 1*

- $i \in \{j, \dots, N+1\} \implies B^{\mathcal{N}}_{1:i}$ remains constant.

**Lemma 7.6** (Increasing a p-value)**.** *Let $p'$ be such that $\|p - p'\|_0 \le 1$. If $p'$ is the increase of a single p-value in $p$ from $B_j$ to $B_{j+s}$, where $j \in \{1, \dots, N\}$ and $1 \le s \le N+1-j$, then*

- $i \in \{1, \dots, j-1\} \implies B^{\mathcal{N}}_{1:i}$ remains constant

- $i \in \{j, \dots, j+s-1\} \implies B^{\mathcal{N}}_{1:i}$ decreases by 1

- $i \in \{j+s, \dots, N+1\} \implies B^{\mathcal{N}}_{1:i}$ remains constant.

**Theorem 7.7** (MOVE-1)**.** *Let $q \in (0,1)$, p-values $\{p_i\}_{i \in \mathcal{N}}$ and sets $\mathcal{H}_0$, $\mathcal{H}_1$ be given. For each $i \in \mathcal{L} \cup \mathcal{R}$, let*

$$FDP_i := \max_{p' : \|p-p'\|_0 \le 1, \tilde{k}'=i} FDP[BH_q; p'].$$

*Then*

$$\max_{p' : \|p-p'\|_0 \le 1} FDP[BH_q; p'] = \left( \max_{i \in \mathcal{L} \cup \mathcal{R} \cup \{\tilde{k}\}} FDP_i \right)$$

This result explains that with one pass of the p-values from the largest to the smallest in the collection, we can ascertain the optimal perturbation $p'$. As the execution based on this result is straightforward, we omit the pseudocode for the sake of brevity.

In Table 3, we compare the average performance of INCREASE-1 against that of the optimal MOVE-1 over $10^4$ simulations. The experiments followed the setup described in Remark 4.6, in which $N = 10^3$ p-values are derived from independently generated z-scores, with $N_0 (= 900)$ null z-scores i.i.d. $N(0,1)$ and $N_1 (= 100)$ alternative z-scores i.i.d. $N(\mu_1, 1)$, for $\mu_1 = 1, 2$. As Table 3 indicates, INCREASE-1 can provide nearly identical performance in adjustment to FDR; however, the perturbation distance $\|z - z'\|$ is on the average much greater than in MOVE-1.

| | MOVE-1(INCREASE-1) | |
|---|---|---|
| | $\mathbb{E}_z FDP[BH_q; z']$ | Average $\|z - z'\|_1$ |
| $\mu_1 = 2$ | 0.140 (0.139) | 0.139 (1.563) |
| $\mu_1 = 1$ | 0.775 (0.751) | 0.492 (2.297) |
| $\mu_1 = 0$ | .992 (0.990) | 0.551 (2.41) |

Table 3: Sample Average ($10^4$-batch) estimates of $\mathbb{E}_z FDP[BH_q; z']$ under MOVE-1 and INCREASE-1 (in parentheses) when $\mu_1 = 0, 1, 2$ and $N = 10^3$, $q = 0.10$, $\pi_0 = 0.90$, and all $\sigma^i = 1$.

## 7.3 Theoretical Analysis: Performance Guarantees and Insights into Adversarial Robustness

**Theorem 4.1.** *If $c \geq 1$, then*

$$\Delta_c \leq \mathbb{P}\left(B^1_{1:c} = N_1\right) \mathbb{E}\left[\frac{c}{c + N_1 + B^0_{1:N_1+c}} \| B^0_{N+1} \geq c\right] + 1 - \mathbb{P}\left(B^1_{1:c} = N_1\right)$$

*and*

$$\mathbb{E}\left[\tilde{k}_{+c} \| B^0_{N+1} \geq c\right] \geq \frac{(N_1 + c) \cdot \mathbb{P}\left(B^1_{1:c} = N_1\right)}{1 - \mathbb{E}\left[\frac{B^0_{1:N}}{N} \| B^0_{1:N} \leq N_0 - c\right]}. \tag{6}$$

*Proof.* When $B^1_{1:c} = N_1$, it easily follows that $\tilde{k}_{+c} \geq c + N_1 + B^0_{1:N_1+c}$. This combined with Theorem 3.2 easily yields the first inequality.

As for the second inequality, let

$$\tilde{k}^0_{+c} := \max\{i \in [0, N]_{\mathbb{Z}} : B^0_{1:i} = i - (N_1 + c)\} \geq N_1 + c > 0.$$

Then

$$\mathbb{E}\left[\frac{B^0_{1:N}}{N} \;\|\; B^0_{N+1} \geq c\right] = \mathbb{E}\left[\frac{B^0_{1:\tilde{k}^0_{+c}}}{\tilde{k}^0_{+c}} \;\|\; B^0_{N+1} \geq c\right] = \mathbb{E}\left[\frac{\tilde{k}^0_{+c} - (N_1 + c)}{\tilde{k}^0_{+c}} \;\|\; B^0_{N+1} \geq c\right] = 1 - (N_1 + c)\mathbb{E}\left[\frac{1}{\tilde{k}^0_{+c}} \;\|\; B^0_{N+1} \geq c\right]$$

implies that

$$\frac{1}{\mathbb{E}\left[\tilde{k}^0_{+c} \;\|\; B^0_{N+1} \geq c\right]} \leq \mathbb{E}\left[\frac{1}{\tilde{k}^0_{+c}} \;\|\; B^0_{N+1} \geq c\right] = \frac{1}{N_1 + c} \cdot \left(1 - \mathbb{E}\left[\frac{B^0_{1:N}}{N} \;\|\; B^0_{N+1} \geq c\right]\right),$$

so that

$$\mathbb{E}\left[\tilde{k}_{+c} \;\|\; B^0_{N+1} \geq c, B^1_{1:c} = N_1\right] = \mathbb{E}\left[\tilde{k}^0_{+c} \;\|\; B^0_{N+1} \geq c\right] \geq \frac{N_1 + c}{1 - \mathbb{E}\left[\frac{B^0_{1:N}}{N} \;\|\; B^0_{N+1} \geq c\right]},$$

since whenever $B^1_{1:c} = N_1$, it follows that $\tilde{k}_{+c} = \tilde{k}^0_{+c}$.

Consequently, we derive

$$\frac{N_1 + c}{1 - \mathbb{E}\left[\frac{B^0_{1:N}}{N} \;\|\; B^0_{N+1} \geq c\right]} \leq \mathbb{E}\left[\tilde{k}_{+c} \;\|\; B^0_{N+1} \geq c, B^1_{1:c} = N_1\right]$$

$$= \frac{\mathbb{E}\left[\tilde{k}_{+c}; B^1_{1:c} = N_1, B^0_{N+1} \geq c\right]}{\mathbb{P}\left(B^1_{1:c} = N_1\right)\mathbb{P}\left(B^0_{N+1} \geq c\right)}$$

$$= \frac{\mathbb{E}\left[\tilde{k}_{+c}; B^0_{N+1} \geq c\right] - \mathbb{E}\left[\tilde{k}_{+c}; B^1_{1:c} < N_1, B^0_{N+1} \geq c\right]}{\mathbb{P}\left(B^1_{1:c} = N_1\right)\mathbb{P}\left(B^0_{N+1} \geq c\right)}$$

$$= \mathbb{E}\left[\tilde{k}_{+c} \;\|\; B^0_{N+1} \geq c\right]\frac{1}{\mathbb{P}\left(B^1_{1:c} = N_1\right)} - \mathbb{E}\left[\tilde{k}_{+c} \;\|\; B^1_{1:c} < N_1, B^0_{N+1} \geq c\right]\frac{\mathbb{P}\left(B^1_{1:c} < N_1\right)}{\mathbb{P}\left(B^1_{1:c} = N_1\right)},$$

yielding

$$\mathbb{P}\left(B^1_{1:c} = N_1\right) \cdot \frac{N_1 + c}{1 - \mathbb{E}\left[\frac{B^0_{1:N}}{N} \;\|\; B^0_{N+1} \geq c\right]} + \mathbb{E}\left[\tilde{k}_{+c} \;\|\; B^1_{1:c} < N_1, B^0_{N+1} \geq c\right] \cdot \mathbb{P}\left(B^1_{1:c} < N_1\right) \leq \mathbb{E}\left[\tilde{k}_{+c} \;\|\; B^0_{N+1} \geq c\right].$$

$\square$

**Corollary 4.2.** *Let $n \geq 1$, $p \in [0, 1]$, and $x$ a non-negative integer such that $0 \leq x \leq n$. If $\tilde{B} = (\tilde{B}_1, \ldots, \tilde{B}_n) \sim Multinomial\left(x, (p, \ldots, p)\right)$, then $\mathbb{P}_{\tilde{B}}\left(\cap^n_{r=1}\left[\sum^r_{i=1}\tilde{B}_i < r\right]\right) = 1 - \frac{x}{n}$.*

*Proof.* We will appeal to the following result of [31]:

**Lemma 7.8** (Theorem 1 of [31]). *Let there be $n \geq 1$ non-negative integers $z_1, \ldots, z_n$ summing to $x$. If $\tilde{\pi}$ denotes a permutation drawn uniformly at random, then*

$$\mathbb{P}_{\tilde{\pi}}\left(\cap_{r=1}^n \left[\sum_{i=1}^r z_{\tilde{\pi}(i)} < r\right]\right) = \left[1 - \frac{x}{n}\right]^+.$$

Let $\tilde{\pi}$ be a random permutation on $\{1, \ldots, n\}$ that is drawn uniformly at random, independently of $\tilde{B}$. Since $\tilde{B} = (\tilde{B}_1, \ldots, \tilde{B}_n) \sim Multinomial\left(x, (p, \ldots, p)\right)$, then $\tilde{B}_{\tilde{\pi}} := (\tilde{B}_{\tilde{\pi}(1)}, \ldots, \tilde{B}_{\tilde{\pi}(n)})$ is equivalent in distribution to $\tilde{B}$, or $\tilde{B}_{\tilde{\pi}} \overset{D}{=} \tilde{B}$. Therefore,

$$\mathbb{P}_{\tilde{B}}\left(\cap_{r=1}^n\left[\sum_{i=1}^r \tilde{B}_i < r\right]\right) = \mathbb{E}_{\tilde{B}}\left[\mathbb{P}_{\tilde{\pi}}\left(\cap_{r=1}^n\left[\sum_{i=1}^r \tilde{B}_{\tilde{\pi}(i)} < r\right]\right)\right]$$

$$= \mathbb{E}_{\tilde{B}}\left[1 - \frac{x}{n}\right] = 1 - \frac{x}{n}.$$

$\square$

**Corollary 4.3.** *If $\mathbb{P}_i^1 = U(0,1)$ for all $i \in \mathcal{H}_1$,*
*then $\mathbb{P}\left(\tilde{k} = \ell\right) = \left(\frac{1-q}{1-q+\frac{(N-\ell)q}{N}}\right) \cdot \mathbb{P}\left(B_{1:\ell}^{\mathcal{N}} = \ell\right)$ for $\ell = 0, \ldots, N$, where $0^0 = 1$,*
*and $\mathbb{P}\left(B_{1:\ell}^{\mathcal{N}} = \ell\right) = \binom{N}{\ell}\left(\frac{q\ell}{N}\right)^\ell\left(1 - \frac{q\ell}{N}\right)^{N-\ell}$.*
*If $B_{N+1}^0 \geq c$, then $\mathbb{P}\left(\tilde{k}_{+c} = c \mid\mid B_{N+1}^0\right) = (1 - \frac{cq}{N})^{N_1} \cdot (1 - \frac{c}{N})^{N_0 - B_{N+1}^0}\left(1 - \frac{N_0 - B_{N+1}^0}{N-c} - \frac{N_1 q}{N-cq}\right).$*

*Proof.* The event $[\tilde{k} = \ell]$ is equivalent to $\{\cap_{j=\ell+1}^N [B_{\ell+1:j}^{\mathcal{N}} < j - \ell]\} \cap [B_{1:\ell}^{\mathcal{N}} = \ell]$, an intersection of two events. Regarding the event $[B_{1:\ell}^{\mathcal{N}} = \ell]$, because $\mu_1 = 0$, it is clear that $\mathbb{P}(B_{1:\ell}^{\mathcal{N}} = \ell) = \binom{N}{\ell}\left(\frac{q\ell}{N}\right)^\ell\left(1 - \frac{q\ell}{N}\right)^{N-\ell}$. To complete the characterization of $\mathbb{P}\left(\tilde{k} = \ell\right)$ it suffices to find $\mathbb{P}\left(\cap_{j=\ell+1}^N [B_{\ell+1:j}^{\mathcal{N}} < j - \ell] \mid\mid [B_{1:\ell}^{\mathcal{N}} = \ell]\right)$. Towards doing so, we note that conditioned on $[B_{1:\ell}^{\mathcal{N}} = \ell]$, we have

$$B_{\ell+1:N}^{\mathcal{N}} \sim Binom\left(N - \ell, \frac{\frac{(N-\ell)q}{N}}{1 - q + \frac{(N-\ell)q}{N}}\right),$$

and, upon further conditioning on $B_{\ell+1:N}^{\mathcal{N}}$,

$$(B_{\ell+1}^{\mathcal{N}}, \ldots, B_N^{\mathcal{N}}) \sim Multinom\left(B_{\ell+1:N}^{\mathcal{N}}, \left(\frac{1}{N-\ell}, \ldots, \frac{1}{N-\ell}\right)\right).$$

Hence, we derive

$$\mathbb{P}\left(\cap_{j=\ell+1}^N [B_{\ell+1:j}^{\mathcal{N}} < j - \ell] \mid\mid [B_{1:\ell}^{\mathcal{N}} = \ell], B_{\ell+1:N}^{\mathcal{N}}\right) = 1 - \frac{B_{\ell+1:N}^{\mathcal{N}}}{N - \ell},$$

by Corollary 4.2. To conclude, we integrate out $B_{\ell+1:N}^{\mathcal{N}}$ from this derivation; more precisely,

$$\mathbb{P}\left(\cap_{j=\ell+1}^N [B_{\ell+1:j}^{\mathcal{N}} < j - \ell] \mid\mid [B_{1:\ell}^{\mathcal{N}} = \ell]\right)$$

$$= \mathbb{E}_{B_{\ell+1:N}^{\mathcal{N}}}\left[\mathbb{P}\left(\cap_{j=\ell+1}^N [B_{\ell+1:j}^{\mathcal{N}} < j - \ell] \mid\mid [B_{1:\ell}^{\mathcal{N}} = \ell], B_{\ell+1:N}^{\mathcal{N}}\right) \mid\mid [B_{1:\ell}^{\mathcal{N}} = \ell]\right]$$

$$= 1 - \frac{\mathbb{E}\left[B_{\ell+1:N}^{\mathcal{N}} \mid\mid [B_{1:\ell}^{\mathcal{N}} = \ell]\right]}{N - \ell} = 1 - \frac{\frac{(N-\ell)q}{N}}{1 - q + \frac{(N-\ell)q}{N}} = \frac{1 - q}{1 - q + \frac{(N-\ell)q}{N}}.$$

As for the second statement, in similar fashion to above, Corollary 4.2 yields

$$\mathbb{P}\left(\cap_{j=c+1}^N [B_{c+1:j}^{\mathcal{N}} < j - c] \mid\mid [B_{1:c}^{\mathcal{N}} = 0], B_{N+1}^0, B_{c+1:N}^1\right) = 1 - \frac{N_0 - B_{N+1}^0 + B_{c+1:N}^1}{N - c},$$

implying

$$\mathbb{P}\left(\cap_{j=c+1}^N [B_{c+1:j}^{\mathcal{N}} < j - c] \mid\mid [B_{1:c}^{\mathcal{N}} = 0], B_{N+1}^0\right) = 1 - \frac{N_0 - B_{N+1}^0 + E[B_{c+1:N}^1 \mid\mid B_{1:c}^1 = 0]}{N - c}.$$

Hence,

$$\mathbb{P}\left(\tilde{k}_{+c} = c \ \| \ B_{N+1}^0\right) = \mathbb{P}\left(B_{1:c}^{\mathcal{N}} = 0 \ \| \ B_{N+1}^0\right) \mathbb{P}\left(\cap_{j=c+1}^{N}[B_{c+1:j}^{\mathcal{N}} < j - 1] \ \| \ B_{1:c}^{\mathcal{N}} = 0, B_{N+1}^0\right)$$

$$= (1 - \frac{cq}{N})^{N_1} \cdot (1 - \frac{c}{N})^{N_0 - B_{N+1}^0}\left(1 - \frac{N_0 - B_{N+1}^0}{N - c} - \frac{N_1 q}{N - cq}\right).$$

$\square$

**Theorem 4.5.** *Suppose Assumption 4.4. Then*

$$\Delta_c \geq L_c(q, N, N_0, \mathbb{P}^1) := \left(1 - \frac{cq}{N} - \delta_{1:c}\right)^{N_1}\left[\left(1 - \frac{c}{N}\right)^{N_0}\left(1 - \pi_c - V_c\right)M_c + Z_c\right]$$

*where* $\pi_c(q, N, N_0, \mathbb{P}^1) := \frac{N_0 + \mathbb{E}[B_{c+1:N}^1 \| B_{1:c}^1 = 0]}{N - c}$, $\quad \mathbb{E}\left[B_{c+1:N}^1 \| B_{1:c}^1 = 0\right] = N_1 \frac{(N-c)q + N\delta_{c+1:N}}{N - cq - N\delta_{1:c}}$,

$V_c(q, N, N_0, \mathbb{P}^1) := \sqrt{\frac{ln2}{2}\mathbb{E}\left[B_{c+1:N}^1 \| B_{1:c}^1 = 0\right]D_{KL}(\mathbb{P}^1, c)}$,

$M_c(q, N, N_0) := \frac{N - cq}{N - c} - \mathbb{E}\left[\left(1 - \frac{c}{N}\right)^{-B_{N+1}^0}; B_{N+1}^0 \leq c - 1\right]$,

$Z_c(q, N, N_0) := \mathbb{P}\left(B_{N+1}^0 \geq c\right)\left(1 - \frac{c}{N}\right)^{N_0 - \mathbb{E}[B_{N+1}^0 \| B_{N+1}^0 \geq c]} \frac{\mathbb{E}[B_{N+1}^0 \| B_{N+1}^0 \geq c]}{N - c}$

$D_{KL}(\mathbb{P}^1, c) := \sum_j \frac{1}{N - c}log\left(\frac{q + \delta - \sum_{j=1}^{c}(q/N + \delta_j)}{(N-c)(q/N + \delta_j)}\right)$

*Proof.* Observe that $\Delta_c = \mathbb{E}\left[\frac{c}{\tilde{k}_{+c}}; B_{N+1}^0 \geq c\right] \geq \mathbb{P}\left(\tilde{k}_{+c} = c, B_{N+1}^0 \geq c\right).$

By Theorem 4.3,

$$\mathbb{P}\left(\tilde{k}_{+c} = c, B_{N+1}^0 \geq c\right) = \mathbb{E}\left[\mathbb{P}\left(\tilde{k}_{+c} = c \ \| \ B_{N+1}^0\right); B_{N+1}^0 \geq c\right]$$

$$= \mathbb{E}\left[\mathbb{P}\left(B_{1:c} = 0 \ \| \ B_{N+1}^0\right) \cdot \underbrace{\mathbb{P}\left(\cap_{j=c+1}^{N}[B_{c+1:j}^{\mathcal{N}} < j - c] \ \| \ [B_{1:c}^{\mathcal{N}} = 0], B_{N+1}^0\right)}_{(\#)}; B_{N+1}^0 \geq c\right]$$

where we now proceed to lower bound (#). The following discussion outlines how we proceed in the case of $c = 1$, but this is without loss of generality.

Let $B_{N+1}^0$ and $B_{2:N}^1$ be given, along with the event $[B_1^{\mathcal{N}} = 0]$. Then there are $N_0 - B_{N+1}^0$ many null p-values and $B_{2:N}^1$ many alternative p-values each of whose assignment to one of bin number $2, \ldots, N$ remains stochastic. Let there be an arbitrary enumeration of these null p-values, upon which we let the collection of their bin-assignment random variables be denoted $(\alpha_i^0)_{i=1}^{N_0 - B_{N+1}^0}$. Let there also be an arbitrary enumeration of these alternative p-values, upon which we let the collection of their bin-assignment random variables be denoted $(\alpha_i^1)_{i=1}^{B_{2:N}^1}$. More precisely, $\alpha_i^0 \sim Unif(\{2, \ldots, N\})$, and $\alpha_i^1 = j$ with probability $\frac{q/N + \delta_j}{q + \delta - (q/N + \delta_1)}$, for $j = 2, \ldots, N$. Finally, we will write $\mathbb{P}_1^{bins}$ for the joint distribution derived from the independent coupling of all the random variables $(\alpha_i^0)_{i=1}^{N_0 - B_{N+1}^0}$ and $(\alpha_i^1)_{i=1}^{B_{2:N}^1}$. For contrast, let $\tilde{\alpha}_i^1 \sim Unif(\{2, \ldots, N\})$, which expresses the random bin assignment of any alternative p-value to one of $B_2, \ldots, B_N$ given that $B_1^{\mathcal{N}} = 0$, were $\mathbb{P}^1 = U(0, 1)$. Then $\mathbb{P}_0^{bins}$, the joint derived from the independent coupling of $(\alpha_i^0)_{i=1}^{N_0 - B_{N+1}^0}$ and $(\tilde{\alpha}_i^1)_{i=1}^{B_{2:N}^1}$, offers a useful counterpoint to $\mathbb{P}_1^{bins}$. By the chain rule for KL-divergence, we find

$$D_{KL}\left(\mathbb{P}_0^{bins}\|\mathbb{P}_1^{bins}\right) = \sum_{i=1}^{N_0 - B_{N+1}^0} D_{KL}\left(\alpha_i^0 \| \alpha_i^0\right) + \sum_{i=1}^{B_{2:N}^1} D_{KL}\left(\tilde{\alpha}_i^1 \| \alpha_i^1\right)$$

$$= B_{2:N}^1 \underbrace{\sum_j \frac{1}{N-1}log\left(\frac{\frac{1}{N-1}}{\frac{q/N + \delta_j}{q + \delta - (q/N + \delta_1)}}\right)}_{=:D_{KL}(\mathbb{P}^1, 1)},$$

and by Pinsker's Inequality, for any event $E$,

$$\mathbb{P}_1^{bins}(E) \geq \mathbb{P}_0^{bins}(E) - \sqrt{\frac{\ln 2}{2} D_{KL}\left(\mathbb{P}_0^{bins} \| \mathbb{P}_1^{bins}\right)}.$$

Let $a_i^0$ (respectively $a_i^1$) denote the realization of the $i$-th null (respectively alternative) p-value's bin assignment. Then if we set $E$ to be the collection of realizations $(a_i^0)_{i=1}^{N_0-B_{N+1}^0}$ and $(a_i^1)_{i=1}^{B_{2:N}^1}$ that are consistent with $\cap_{j=2}^N[B_{2:j} < j-1]$, we find

$$\mathbb{P}\left(\cap_{j=2}^N[B_{2:j}^{\mathcal{N}} < j-1] \ \| \ B_1^{\mathcal{N}} = 0, B_{N+1}^0, B_{2:N}^1\right)$$

$$= \mathbb{P}_{\mu_1}(E) \geq \mathbb{P}_0(E) - \sqrt{\frac{\ln 2}{2} D_{KL}\left(\mathbb{P}_0^{bins} \| \mathbb{P}_1^{bins}\right)}$$

$$= \left(1 - \frac{N_0 - B_{N+1}^0 + B_{2:N}^1}{N-1}\right) - \sqrt{\frac{\ln 2}{2} B_{2:N}^1 D_{KL}(\mathbb{P}^1, 1)}. \qquad \text{(Corollary 4.2)}$$

Hence, for the case of general $c$,

$$(\#) = \mathbb{E}\left[\mathbb{P}\left(\cap_{j=c+1}^N[B_{c+1:j}^{\mathcal{N}} < j-c] \ \| \ [B_{1:c}^{\mathcal{N}} = 0], B_{N+1}^0, B_{c+1:N}^1\right) \ \| \ [B_{1:c}^{\mathcal{N}} = 0], B_{N+1}^0\right]$$

$$\geq \mathbb{E}\left[\left(1 - \frac{N_0 - B_{N+1}^0 + B_{c+1:N}^1}{N-c}\right) - \sqrt{\frac{\ln 2}{2} B_{c+1:N}^1 D_{KL}(\mathbb{P}^1, c)} \ \| \ [B_{1:c}^{\mathcal{N}} = 0], B_{N+1}^0\right]$$

$$\text{(Pinsker's Inequality)}$$

$$\geq 1 - \frac{N_0 - B_{N+1}^0 + \mathbb{E}\left[B_{c+1:N}^1 \| B_{1:c}^1 = 0\right]}{N-c} - \sqrt{\frac{\ln 2}{2} \mathbb{E}\left[B_{c+1:N}^1 \| B_{1:c}^1 = 0\right] D_{KL}(\mathbb{P}^1, c)}.$$

It follows that

$$\mathbb{P}\left(\tilde{k}_{+c} = c, B_{N+1}^0 \geq c\right) \geq$$

$$\mathbb{E}\left[\underbrace{\mathbb{P}\left(B_{1:c}^{\mathcal{N}} = 0 \ \| \ B_{N+1}^0\right)}_{(1-\frac{cq}{N}-\delta_{1:c})^{N_1}(1-\frac{c}{N})^{N_0-B_{N+1}^0}}\left(1 - \frac{N_0 - B_{N+1}^0 + \mathbb{E}\left[B_{c+1:N}^1 \| B_1^1 = 0\right]}{N-c} - \sqrt{\frac{\ln 2}{2} \mathbb{E}\left[B_{c+1:N}^1 \| B_1^1 = 0\right] D_{KL}(\mathbb{P}^1, c)}\right); B_{N+1}^0 \geq c\right].$$

To conclude,

$$\frac{\mathbb{P}\left(\tilde{k}_{+c} = c, B_{N+1}^0 \geq 1\right)}{(1 - \frac{cq}{N} - \delta_{1:c})^{N_1}}$$

$$\geq \mathbb{E}\left[(1-\frac{c}{N})^{N_0-B_{N+1}^0}\left(1 - \frac{N_0 - B_{N+1}^0 + \mathbb{E}\left[B_{c+1:N}^1 \| B_{1:c}^1 = 0\right]}{N-c} - \sqrt{\frac{\ln 2}{2} \mathbb{E}\left[B_{2:N}^1 \| B_1^1 = 0\right] D_{KL}(\mathbb{P}^1, c)}\right); B_{N+1}^0 \geq c\right]$$

$$= \left(1-\frac{c}{N}\right)^{N_0}\left(1 - \underbrace{\frac{N_0 + \mathbb{E}\left[B_{c+1:N}^1 \| B_{1:c}^1 = 0\right]}{N-c}}_{\pi(c,\mu_1)} - \underbrace{\sqrt{\frac{\ln 2}{2} \mathbb{E}\left[B_{c+1:N}^1 \| B_{1:c}^1 = 0\right] D_{KL}(\mathbb{P}^1, c)}}_{V(c,\mu_1)}\right)\underbrace{\mathbb{E}\left[(1-\frac{c}{N})^{-B_{N+1}^0}; B_{N+1}^0 \geq c\right]}_{(**)}$$

$$+ \underbrace{\mathbb{E}\left[(1-\frac{c}{N})^{N_0-B_{N+1}^0}\frac{B_{N+1}^0}{N-c}; B_{N+1}^0 \geq c\right]}_{(***)},$$

where

$$(**) = \mathbb{E}\left[\left(1-\frac{c}{N}\right)^{-B_{N+1}^0}\right] - \mathbb{E}\left[\left(1-\frac{c}{N}\right)^{-B_{N+1}^0}; B_{N+1}^0 \leq c-1\right] = \frac{N-cq}{N-c} - \mathbb{E}\left[\left(1-\frac{c}{N}\right)^{-B_{N+1}^0}; B_{N+1}^0 \leq c-1\right],$$

$$(***) \geq \mathbb{P}\left(B_{N+1}^0 \geq c\right)\left(1-\frac{c}{N}\right)^{N_0-\mathbb{E}[B_{N+1}^0 \| B_{N+1}^0 \geq c]}\frac{\mathbb{E}\left[B_{N+1}^0 \| B_{N+1}^0 \geq c\right]}{N-c} \qquad \text{(Jensen's)}$$

$$\square$$

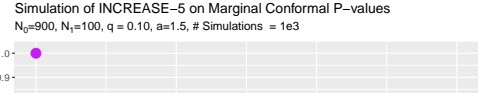

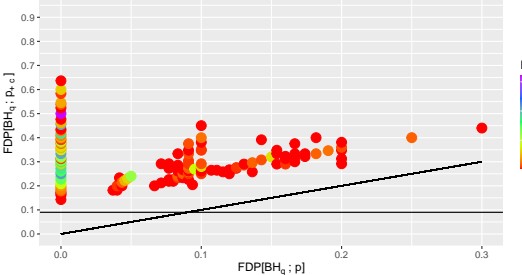

| $c$ | $a$ | 1 | 1.5 | 2 | 2.5 | 3 |
|---|---|---|---|---|---|---|
| 1 | | 1 | .559 | .096 | .096 | .098 |
| 5 | | 1 | .368 | .142 | .135 | .133 |
| 10 | | .997 | .435 | .190 | .174 | .170 |
| 50 | | .992 | .657 | .429 | .389 | .383 |

Figure 4: $10^3$ simulations of FDP by $BH_q$ with and without application of INCREASE-5 on marginal conformal p-values [5] derived from an SVM one-class classifier on a test set with outliers drawn with $a = 1.5$.

Table 4: Sample Average ($10^3$-batch) estimates of $\mathbb{E}FDP[BH_q; p_{+c}]$ on marginal conformal p-values for outlier detection [5], in terms of signal strength $a$ and budget $c$.

## 7.4 Simulations and Data Experiments

### 7.4.1 INCREASE-c Simulations on PRDS p-values

In Table 4 we illustrate the effectiveness of INCREASE-c in disrupting the nonparametric outlier detection method of [5] that is based on the application of BH on conformal p-values, and in doing so, demonstrate effectiveness of INCREASE-c on PRDS p-values. We follow the simulation setting of Section 5.2 in [5], using their publicly available source code to generate the conformal p-values. In short, a data set is generated in $\mathbb{R}^{50}$, along with $10^3$ training observations used to fit a one-class SVM classifier, as well as $10^3$ observations forming a calibration set to be used with a test set to derive (marginal) conformal p-values. In each of $10^3$ independent replications, INCREASE-c was applied to a new test set consisting of $10^3$ conformal p-values, designed to discern inliers (signals drawn from a mixture of multivariate gaussians with identity covariance matrices) from outliers (signals drawn from a mixture of multivariate gaussians with identity covariance matrices scaled by a strength $\sqrt{a}$). This was performed for $a \in \{1, 1.5, 2, 2.5, 3\}$; a signal strength $a = 1$ corresponds to identical null and alternative distributions, while larger values of $a$ make it easier to detect outliers. We set the fraction of outliers in each test set to $\pi_1 = .1$, so that a fraction $\pi_0 = .90$ of the observations are inliers in each data set.

