# OpenReview forum: "On the Adversarial Robustness of Benjamini Hochberg"
_NeurIPS.cc/2024/Conference — NeurIPS 2024 poster_

### Official Review · Reviewer_niTJ · 2024-07-10

**Soundness:** 3
**Presentation:** 2
**Contribution:** 2
**Rating:** 6
**Confidence:** 3

**Summary:**

Benjamini-Hochberg provides a means for controlling the false detection rate (FDR) in multiple hypothesis testing.  This paper explores how the FDR region can be perturbed by an adversary.  Specifically, the author(s) propose two adversarial attacks *Increase-C* and *MOVE-1* to efficiently perturb *c* scores and 1 score, respectively.  The author(s) prove analytical bounds on their attacks' guarantees.  The author(s) also provide simulated experiments demonstrating their attacks effectiveness on synthetic z-scores.

**Strengths:**

I was not familiar with Benjamini-Hochberg before reading this paper (I did not bid on the paper).  However, with significant effort and outside reading, the paper was still attainable.  That is a strength of the work in general.  The *Balls in Bins* warm-up was definitely helpful to a reader.  I provide comments on how this warm-up could be improved below.

### Novel Area and Good Execution

To the extent of my knowledge, no existing work has studied the adversarial robustness of methods to control FDR.  I discuss why I believe that is below in the limitations section.  Nonetheless, the authors provide reasonable first-pass attack methods that are intuitive and performant.  They even provide an optimal method in the case of singleton perturbations.  The authors also provide thorough theoretical analysis.

**Weaknesses:**

### Very Little Analysis of Section 5's Empirical Results

Sections 5.1 and 5.2 are essentially just an overview of the two experimental setups.  There is no real analysis or discussion of what the experimental results show or why it's important.  While Section 5.3 has some basic analysis, it is again quite minimal.  I understand that the page limit forces difficult decisions over content but so little in the way of commentary on the results is a disservice to the reader and overall a poor choice.

I would have preferred empirical results on real datasets as opposed to the toy datasets in the paper, but I understand the author(s)'s choices here, especially given related work.

### Move-1 Relegated to the Appendix

Beyond a very definition in Section 1.5, all discussion of MOVE-1 including how the method works and why its optimal are relegated to the appendix.  There should at least a basic overview of the key ideas of MOVE-1 including some basic intuitions how it differs from INCREASE-c in the main paper.

### Making the Work More Intuitive

Section 2 in my view is a warmup for readers to help build intuitions and understanding of BH, even if they are not familiar with the related background. Section 2 could be significantly improved if figures were used to better illustrate the key ideas.  These figures need not be large and any additional camera ready page could be used to visualize the ideas.  Even if the authors chose not to include such figures in the main paper, they could be in the appendix.

### Limited Potential Impact

There is undeniable merit and utility in studying the robustness of a method that provide tools to control FDR, such as Benjamini-Hochberg.  Nonetheless, this work falls essentially is a niche (BH) within a niche (adversarial analysis).  Hence, I view it as unlikely that this work will have a significant impact.

**Questions:**

None.

**Limitations:**

This paper describes two adversarial attacks and identifies the brittleness of Benjamini-Hochberg.  While such work inherently creates a non-zero risk, there are no serious negative societal impact.

---

> ### Author Rebuttal · Authors · 2024-08-06
>
> Addressing the *``Weaknesses"* $\ldots$
>
> *1. Very Little Analysis of Section 5's Empirical Results*
>
>    *Sections 5.1 and 5.2 are essentially just an overview of the two experimental setups. There is no real analysis or discussion of what the experimental results show or why it's important. While Section 5.3 has some basic analysis, it is again quite minimal. I understand that the page limit forces difficult decisions over content but so little in the way of commentary on the results is a disservice to the reader and overall a poor choice.*
>
>    **Rebuttal:**
>
>    **In terms of Sections 5.1 and 5.2, which concern simulating INCREASE-c on iid and PRDS p-values respectively, we had thought the figures and tables illustrated the effectiveness of INCREASE-c far better than words could. Indeed, the page limit did severely constrain some discussions, but if given the opportunity to, we would absolutely correct this with more elaboration, as you suggest. For what it's worth, what we had intended for the reader to understand from Figure 1, was that with both colors and lines, we were indicating how frequently INCREASE-c not only increased the FDP (above the $45^\circ$ line) but also raised it above the theoretical control level of $\pi_0 \cdot q$ (above the horizontal line), or in other words, broke BH's FDR control. Likewise in Table 1, the high average FDP numbers and perturbed rejection counts illustrate the effectiveness of INCREASE-c. Figure 4 and Table 2 communicated the similar conclusion but for PRDS conformal p-values.**
>
>    *I would have preferred empirical results on real datasets as opposed to the toy datasets in the paper, but I understand the author(s)'s choices here, especially given related work.*
>
>    **Rebuttal:**
>
>    **Thanks for your input. For us, given the choice between a real dataset and a simulation-procured, synthetic dataset of Bates, Candes et al 2023 -- with publicly available code which they used to conduct their experiments -- we felt our message on BH's fragility would be stronger by repurposing their code for comparison's sake. Consequently, in light of the page limit, we had to compromise on the real datasets.**
>
> *2. Move-1 Relegated to the Appendix*
>
>    *Beyond a very definition in Section 1.5, all discussion of MOVE-1 including how the method works and why its optimal are relegated to the appendix. There should at least a basic overview of the key ideas of MOVE-1 including some basic intuitions how it differs from INCREASE-c in the main paper.*
>
>    **Rebuttal:**
>
>    **We thank you for your interest and appreciation of MOVE-1. Of course we appreciate it too, but it not only pertains to the very special case of $c = 1$ but also does not strongly outperform INCREASE-1 empirically per se, despite its theoretical optimality in this case. Indeed, the INCREASE-c algorithm is arguably easier to describe and implement, and of course handles all possible $c$ value cases. Our theoretical performance analysis guarantee was also designed specifically for INCREASE-c. Hence, for all these reasons, and in light of the 9 page limit, we did not feel as though discussing MOVE-1 in detail had good bang for buck.**
>
> *3. Making the Work More Intuitive*
>
>    *Section 2 in my view is a warmup for readers to help build intuitions and understanding of BH, even if they are not familiar with the related background. Section 2 could be significantly improved if figures were used to better illustrate the key ideas. These figures need not be large and any additional camera-ready page could be used to visualize the ideas. Even if the authors chose not to include such figures in the main paper, they could be in the appendix.*
>
>    **Rebuttal:**
>
>    **Your suggestions to use illustrations to enhance the readability of the mathematical description of the Balls into Bins perspective on BH are duly noted. Thank you!**
>
> *4. Limited Potential Impact*
>
>    *There is undeniable merit and utility in studying the robustness of a method that provides tools to control FDR, such as Benjamini-Hochberg. Nonetheless, this work falls essentially is a niche (BH) within a niche (adversarial analysis). Hence, I view it as unlikely that this work will have a significant impact.*
>
>    **Rebuttal:**
>
>    **Thank you for your assessment. The BH is the tool of choice for large scale hypothesis testing, with over 100K Google scholar citations of the original Benjamini and Hochberg "Controlling the false discovery rate: a practical and powerful approach to multiple testing" paper. As well, in the intro to our paper, we also cite a handful of more modern works in AI/ML that are leveraging BH in some way for out-of-distribution-detection (OOD) - among them, Lieu et al 2024 (accepted at the recent ICML 2024) and Bates, Candes et al 2023. So, we believe a paper examining BH's fragility is relevant and important.**
>
> ---

---

### Official Review · Reviewer_3drS · 2024-07-13

**Soundness:** 3
**Presentation:** 2
**Contribution:** 3
**Rating:** 5
**Confidence:** 1

**Summary:**

This paper studies the adversarial robustness of the Benjamini-Hochberg (BH) procedure, introducing simple adversarial test-perturbation algorithms. The experiments show that BH's control can be significantly compromised with minimal perturbations. The analysis uses a combinatorial perspective and generalized ballot problems to derive non-asymptotic lower bounds.

**Strengths:**

This paper addresses a novel and significant aspect of the BH procedure—its adversarial robustness. This research probelm is relatively unexplored, providing new insights into the vulnerabilities of widely used statistical methods.

**Weaknesses:**

This paper would benefit from a discussion of potential mitigation strategies that could enhance the robustness of the Benjamini Hochberg procedure against attacks such as INCREASE-c.

**Questions:**

Please refer to the weaknesses.

**Limitations:**

The authors' justification is "Yes, we discuss to what extent our analysis could be extended and further generalized, but were not done due to space constraints.". However, even in Appendix, there is not any analysis of limitations.

---

> ### Author Rebuttal · Authors · 2024-08-06
>
> Addressing the Weakness comment $\ldots$
>
> *This paper would benefit from a discussion of potential mitigation strategies that could enhance the robustness of the Benjamini Hochberg procedure against attacks such as INCREASE-c.*
>
> **Rebuttal:**
>
> **Thanks for asking about mitigation strategies, which we view as an important follow-up question to this paper's thesis that BH's FDR control can be susceptible to minimal adversarial perturbation. In fact, we are in the midst of preparing this follow-up work. We kindly refer you to our answer to Reviewer pEvv's question #2, in which we briefly comment on interesting dynamics to the attacker-defender interaction, including some insights into mitigation strategies studied in this said follow-up work. Given that our submission as it stands now rides right up to the 9-page limit, with a moderate amount of appendix material already, we believe that the study of mitigation strategy extensions would be better positioned as a separate paper.**

---

### Official Review · Reviewer_L7r5 · 2024-07-26

**Soundness:** 3
**Presentation:** 4
**Contribution:** 3
**Rating:** 5
**Confidence:** 4

**Summary:**

In this paper, the authors have tested the Benjamini-Hochberg (BH) 's adversarial robustness, as this procedure is deployed in critical applications such as drug discovery, forensics, and anomaly detection. Specifically, the authors develop a class of simple and easily implementable adversarial test-perturbation algorithms to analyze under what conditions BH does and does not exhibit adversarial robustness. Next, to support their findings, the authors provide non-asymptotic guarantees on the expected adjustments to the FDR due to these adversarial attacks. Their technical analysis involves a combinatorial reframing of the BH procedure as a "balls into bins" process. It connects this to generalized ballot problems to utilize information-theoretic approaches for deriving non-asymptotic lower bounds. Finally, the authors also conducted experiments to support their findings.

**Strengths:**

- In this paper, the authors provided a detailed theoretical analysis of the BH procedure's robustness against adversarial attacks. They also introduced the INCREASE-c algorithm, which gives methodical mathematical bounds and probabilities for different cases (large and small alternative means).

- I liked how advanced probabilistic and statistical tools, such as KL divergence and Pinsker's inequality, are used to provide a solid mathematical foundation for the analysis. The proofs and the lemmas were thoroughly justified.

- Finally, the paper presentation is well-structured. The paper's organization is clear and logical. In each section, a comprehensive analysis is done and well-presented.

**Weaknesses:**

- In practical scenarios, an adversary manipulates the input directly, not the z-scores; thus, the insight into how perturbations applied to z-scores translate back to the original data samples or vice-versa is missing in the paper.

- My other concern is that since z-scores are computed only utilizing the means and the standard deviation of the sample, the manipulation at the data level can get lost at the z-scores level.

**Questions:**

- In practical scenarios, an adversary manipulates the input directly, not the z-scores; thus, could the authors provide more insight into how perturbations applied to z-scores translate back to the original data samples or vice-versa? Or, when the inputs are perturbed, what will impact the proposed analysis of the BH algorithm (given the z-scores computed from the input samples)?

- A statement on Page 4: "A benefit of this approach is that perturbation at the level of z-scores places the corruption more directly at the point of data collection than does perturbation of the p-values." Could the authors answer: Won't the perturbation at the data collection directly place the corruption at the data level? Also, why was it not assumed to be corruption at the data level, then the computation of either the z-scores or the p-values? How this mapping between the data and the z-scores is computed?

- The evaluation results are a bit difficult to navigate in the paper. For example, Figure 1 and Figure 4 are not understandable. Could the authors make the evaluation section comprehensible?

- Could the authors add more evaluations to test how the proposed method performs with real-world data?


Post-rebuttal: I have changed my rating from borderline reject to borderline accept, confidence from 3 to 4, and soundness of the paper from 2 to 3.

**Limitations:**

Same as Weaknesses.

---

> ### Author Rebuttal · Authors · 2024-08-07
>
> # Q1 Re:
>  Indeed, $z$ scores are classically $z_i := \frac{\sum_{j=1}^n x_{ij}/n - 0}{s/\sqrt{n}}$, so that perturbations to the "samples" $x_{ij}$ translate to perturbation of the z-score $z_i$, and ultimately the p-value $p_i:= 1 - \Phi(z_i)$. Since p-values are ultimately the input processed by BH, our model is not concerned with how perturbations of the input samples $x_{ij}$ are made, so long as the $p_i$ value is affected.
>
> For a broader perspective on this matter of samples versus z-scores versus p-values, we call to mind the application of ``conformal p-values" by Bates, Candes, et al 2023 that strongly motivated this work, and for which we reference several times, and implement their experiments within ours in Section 5.2. In particular, they consider a statistical wrapper that transforms each $X_i$ in a collection $(X_j)_{j=1}^n$
>
> of signals (/measurements) into a p-value $p_i \in [0,1]$, the total collection $\{p_j\}_{j=1}^n$ of which corresponds to $n$ hypothesis tests to be conducted simultaneously, each of which determines whether a signal $X_i$ is an anomaly, or outlier. Therefore, in this context, perturbations to a score $X_i$ is direct.
>
> # Q2 Re:
>
> As far as theoretical analysis, our work is largely free of distributional assumptions, so there is little "impact" from "data-generating distribution changes." For instance, it is w.l.og. that the null p-values are Uniformly distributed. As for the alternative distributions, we made few assumptions throughout. Indeed, Theorems 3.1, 3.2 after all do not make any distributional assumptions on the alternatives. The bound provided in Theorem 4.4 is a function of the $\mu_1^i$, which as we discussed in Section 4.2, can also be replaced with max $\mu_1:= \max_{i \in \mathcal{H}_1} \mu_1^i$ for a more conservative bound.
>
> As far as numerical analysis, it was for the sake of experiments that we made assumptions on the alternative distribution, as do all papers dealing with the BH procedure - see Bates, et al 2023 for example.
>
> # Q3 Re:
>  To the best of our understanding, we address your concern in our answer to Question 1, so we kindly refer you to the answer we provided above.
>
> # Q4 Re:
>  With regards to the Type II error, since all the z-scores will be in the rejection region, there would be no failure in rejecting any $i \in \mathcal{H}_1$ (i.e. any test for which the ``alternate hypothesis is true"); hence, there would be no Type II error, by definition. For what it's worth, although Type II error is of note, false discovery rate control is concerned with Type I error.
>
> # Q5 Re:
> If we're not mistaken, in the "same case above", the alternative distributions' being "close" to the null is a non-factor. Indeed, you provide an event in which all $N$ z-scores are rejected, so the Type II error is 0, as above. As for the FDP in this event, it would be forced to be $\pi_0$ because this is the fraction of scores out of $N$ that are null.
>
> If you are asking about FDR control in the case of alternative distributions being "close" to the null distribution, this is precisely the subject of Section 4.2, wherein we provide a theoretical lower bound (Theorem 4) on how much the adversary can increase the FDR with INCREASE-c. This bound is plotted in Figures 2 and 3 of Section 5, in which we examine, respectively, the extreme case when the alternative distributions are identical to the null distribution and a case where they are not identical but roughly speaking, about a quarter standard deviation away from each other. The point to these plots is to illustrate the fragility of the BH FDR control when the alternative and null distributions are "close".
>
> # Q6 Re:
>  The frequency color of a plotted dot at $(FDP[BH_q; p]$, $FDP[BH_q; p_{+c}]) \in \mathbb{R}^2$ indicates how many of the $10^4$ simulations produced that combination of before-adversary and after-adversary FDP. Figures 1 and 4 illustrate how very frequently the adversary's INCREASE-c not only increased the FDP (above the $45 \deg$ line) but also raised it above the theoretical control level of $\pi_0 \cdot q$ (above the horizontal line). This high frequency illustrates a breaking of BH's FDR control.
>
> # Q7 Re:
>  We believe you are referring to Section 5.1, line 311 which states $z_{i \in \mathcal{N}_0} \sim N(\mu_1, 1)$.
>
> There was a typo in the subscript, and should actually read: $z_{i \in \mathcal{N}_1} \sim N(\mu_1, 1)$.
>
> # Q8 Re:
>  We followed the approach taken in most BH-procedure papers, where the experiments are over a range of $\mu_1$. Unfortunately, not all parameter choices could be explored/presented within a 9-page paper. For what it's worth, follow-up experiments do not seem to indicate anything that we haven't showcased already.
>
> # Q9 Re:
>   We executed the algorithm just as described on page 5. We "move the largest $c$ (ties broken arbitrarily) in the (N+1)-th bin to bin $\tilde{k}_{+c}$." More precisely, as our z-scores were stored in memory with vectors, ties were broken by taking whichever z-score occurred earlier in the vector's indexed-ordering.
>
> # Q10 Re:
>  Section 5.3's experimental setup involved repeated simulations of $N = 10^3$ p-values drawn just as described in Section 5.1. Regarding the experiment that produced Figure 2, as the caption indicates, we had parameter settings of $\mu_1 = 0$, $\pi_0 = .90,$ and $c = 1.$ As for $q$, as the figure indicates, this was now varied across a grid of $q \in (0,1)$ (mesh spacing of .01). For each $q,$ we performed the repeated simulations, in each of which we computed $FDP[BH_q; z_{+c}] - FDP[BH_q; z]$, so that averaged across all simulations we obtained $\Delta_1(q)$. Repeated for all $q \in (0,1)$, we obtained the blue curve that is labeled $\Delta_1$. The red curve is $L_c$ from Theorem 4.4 as a function of $q$. Regarding the experiment that produced Figure 3, as the caption indicates, we had parameter settings of $\mu_1 = 0.25$, $\pi_0 = .95,$ and $c = 1.$ The rest of the details are just as described above.

---

> > ### Comment · Reviewer_L7r5 · 2024-08-09
> >
> > Dear Authors,
> > Thank you for your response. The rebuttal clarified the majority of my questions.
> > - However, there remains an open concern (from all the reviewers) about how well the proposed method would perform with real-world data. Given the page limit of 9 pages, could the authors experiment with an MNIST/FashionMNIST OOD-classifier to demonstrate the efficacy of the INCREASE-c algorithm, as suggested by reviewer pEvv, and add it in the appendix? It will help strengthen the paper.
> >
> > -  This will also show the practical relevance of the proposed approach, as real-world adversarial attacks are more likely to target the raw input data.

---

> > > ### Author Response · Authors · 2024-08-12
> > >
> > > Yes, we would be happy to add experiments with real-world data to the appendix. We have provided an experiment in our "global" official comment above, which serves as an example of things we could provide in the appendix, if given the opportunity to do so.
> > >
> > > Thank you for your suggestion!

---

### Official Review · Reviewer_pEvv · 2024-07-26

**Soundness:** 3
**Presentation:** 4
**Contribution:** 2
**Rating:** 6
**Confidence:** 3

**Summary:**

The paper explores the adversarial robustness of the Benjamini-Hochberg procedure. In particular, the authors theoretically show that it is possible to perturb test scores to cause the BH procedure to not be robust to adversarial attacks. BH is reframed as a "balls into bins" problem, and the authors propose an algorithm (INCREASE-c) to increase the rejection count. The authors also provide experiments to support their theoretical findings.

**Strengths:**

- The technical contribution seems solid and sound.
- There is a clear novelty aspect, as the adversarial robustness of BH has not been considered before.
- The authors show an interesting gap between distributional robustness (which was shown to hold (ref [29] in the paper)) and adversarial robustness for BH procedure.
- The (synthetic) statistical experiment supports the theoretical claims.

**Weaknesses:**

- In the introduction, the paper (correctly) highlights the importance of hypothesis testing to various safety applications such as OOD detection. However, the experimental section does not, in any capacity, consider an end-to-end application such as OOD detection. In fact, it reads purely as a statistical simulation (which is fine as a synthetic experiment). But a minimal experiment with a MNIST/FashionMNIST OOD-classifier, to show that indeed the Z-scores can be affected by INCREASE-c could make the paper much stronger.
- The attack is dependent on the algorithm (BH) used for FDR -- stronger results in adversarial robustness are often model/algorithm-independent.
- There seems to be a certain lack of justification for choices in the problem setup (addressed in questions below).

**Questions:**

- Problem setup questions:
	- What is the rationale for the attacker having access to z-scores at that stage of the pipeline? Usual studies consider attackers having access to data, and it may take quite a lot of changes in data points to change a z-score.
	- Why is the attacker's knowledge of $q$ realistic? I assume that in some models, the deployment will come with a (public) guarantee for $q$ -- but this should be at least mentioned.
	- It would be good to have a comment justifying the choice of perturbation ( $||\cdot||_0$ ), which I suppose is because BH is an algorithm for order statistics. What happens when considering a different type of budget (e.g., arbitrary $||\cdot||_p$ norms)?
- l.128-132: When are these assumptions met in practice?
- What is the impact of attacking BH procedure on the whole OOD system? Can you break OOD detection in a meaningful way?


****

Post-rebuttal: I have increased my confidence score from 2 to 3 following the authors' response and other reviews.

**Limitations:**

I think some of the problem setup questions above would address some limitations of the work.

---

> ### Author Rebuttal · Authors · 2024-08-06
>
> **Addressing Weaknesses**
>
> 1. *In the introduction, the paper (correctly) highlights the importance of hypothesis testing $\ldots$ However, the experimental section does not $\ldots$ consider an application such as OOD detection \ldots*
>
> **Rebuttal:**
> We did in fact experiment on an ``OOD-classifier." We refer you to our Section 5.2's experiments on the non-parametric outlier detection method proposed in the paper by Bates, Candes, et al 2023 based on conformal p-values. Specifically, we repeated the experiment in their paper that produced conformal p-values with a trained, SVM one-class classifier, only now we added small corruption to see how badly their OOD system might mistake inliers for outliers.
>
> 2. *The attack is dependent on the algorithm (BH) used for FDR -- stronger results in adversarial robustness are often model/algorithm-independent.*
>
> **Rebuttal:**
> BH is the tool of choice for large scale hypothesis testing, with over 100K Google scholar citations of the original paper. BH is also emerging in AI/ML for out-of-distribution-detection (OOD) - for example, see Lieu et al 2024 (accepted at the recent ICML 2024) and Bates, Candes et al 2023. In other words, BH would seem to be a natural and relevant choice for an adversarial study.
>
> Further, BH belongs to the family of step-up procedures, which means the rejection region is decided via a stopping time (see our analysis), which since our paper shows can be manipulated, it means other ``step-up" procedures" are similarly prone.
>
> **Addressing Questions**
>
> Question 1. *What is the rationale for the attacker having access to z-scores at that stage of the pipeline? Usual studies consider attackers having access to data, and it may take quite a lot of changes in data points to change a z-score.*
>
> **Rebuttal:**
>
> Thank you for your comment. For what it's worth, changes in the data translate into changes at the z-score level. As for the size of changes, in Page 16, Appendix Table 3, the algorithm MOVE-1 on average moved the z-score .551, .492, and .139 in the settings of $\mu_1 = 0, 1,$ and $2$ respectively. Granted, the effect of data-point perturbations on z-scores would ultimately also be affected by matters like sample size and standard deviation. That being said, we are more focused on applications in outlier detection or out-of-distribution-detection (OOD), as in Bates et al 2023, which we referenced several times and use as an experimental baseline in Section 5.
>
> More precisely, in this context, they consider a statistical wrapper that, broadly speaking, transforms each $X_i$ in a collection
> $X_1, X_2, \ldots X_n$
> of signals (measurements) into a p-value $p_i \in [0,1]$, the total collection $(p_j)_{j=1}^n$ of which then corresponds to $n$ hypothesis tests to be conducted simultaneously, each of which determines whether a signal $X_i$ is an anomaly/outlier. Therefore, perturbations in this context affect the $X_i$ scores directly.
>
> For further discussion on this, we kindly refer you to our answer to reviewer L7r5's question \#1.
>
> Question 2. *Why is the attacker's knowledge of $q$ realistic? $\ldots$*
>
> **Rebuttal:**
>
> Indeed, your point about a "public guarantee" that would grant the adversary knowledge is realistic, with values of $q = 0.05, 0.10$ a standard practice.
>
> On a related note, while it is possible for the adversary to not precisely know the true control level $q$ to be implemented by the decision maker, this opens the door to some really interesting attacker-defender dynamics between the adversary and decision maker that we are in fact studying in a follow-up work. In any case, we confirm that there are measures that can be taken by the adversary towards harming FDR control even without perfect knowledge of $q.$
>
> Question 3. It would be good to have a comment justifying the choice of perturbation $(||\cdot||_0)$,
>
> which I suppose is because BH is an algorithm for order statistics. What happens when considering a different type of budget (e.g., arbitrary $||\cdot ||_{p}$ norms)?
>
> **Rebuttal:**
>
> Viewed from the perspective of outlier detection, our study is considering an adversary attempting to make the decision maker confuse inlier signals for outlier signals or vice versa. For some particular motivating applications, consider candidate screening, spotting frauds/intrusions, and forensic analysis - applications discussed in Bates, Candes, et. al 2023 and Jin, Candes 2023. In such contexts, we think it is natural that the *number* of hypothesis tests (equiv. signals) that an adversary can influence is bounded - hence, the modeling choice of a budgeted $\|\cdot \|_0$. This choice was sufficient to demonstrate a fragility to BH's FDR control. More precisely, the attacker can break the BH guarantee by moving few z-scores. But, yes, other measures ($\|\cdot \|_p$ norms) of corruption effort might be considered as well.
>
> Question 4. *l.128-132: When are these assumptions met in practice?*
>
> **Rebuttal:**
>
> The lines l.128-132 do not detail assumptions; rather, all we mean to say in these lines is that the corruption model we propose is most interesting in scenarios where not too many tests get rejected, for otherwise any attack would have to be massive in order to have an effect on the FDR. Indeed, the number of rejections $R_p$ comprises the denominator in the false detection proportion $FDP[\mathcal{A};p]:= \frac{a_p}{R_p \vee 1}$. Regardless, the theoretical results hold true under any values of $N$, $q$, and the collection of $\mu_1^i$'s.
>
> Question 5. *What is the impact of attacking BH procedure on the whole OOD system?... break OOD detection?*
>
> **Rebuttal:**
>
> Figure 4 and Table 2 detail the experiments we conducted on the OOD "system" that Bates, Candes, et al 2023 studied. In particular, in Table 2, the average proportion of the reported outliers that are in fact inliers is reported. As one can see, there can be significant impact on OOD systems based on BH (of which there are several recent publications on).

---

> > ### Comment · Reviewer_pEvv · 2024-08-12
> > **Rebuttal Response**
> >
> > Dear authors,
> >
> > Thank you for the detailed rebuttal and the clarifications.
> >
> > Regarding the OOD application, my comment was more in line with reviewer L7r5, in that an experiment on real-world data (rather than a synthetic one) would significantly strengthen the paper.
> >
> > Regarding the algorithmic-dependent results, a remark in the spirit of the last sentence of your response to this weakness (*Further, BH belongs to the family of step-up procedures, which means the rejection region is decided via a stopping time (see our analysis), which since our paper shows can be manipulated, it means other ``step-up" procedures" are similarly prone.*) would be worth including in a future version, perhaps along with more general theorem statements/corollaries.
> >
> > My last remaining question for the author is whether they would be willing/able to add justifications/explanations in the paper in line with my and other reviewer's comments and questions on z-scores perturbation, choice of norm, adversary's power, etc. (which would make the paper clearer, a bit more solid and easier to approach for people who are not as familiar with the literature), and whether they have or will perform an experiment on real-world data as suggested by L7r5 and myself.

---

> > > ### Author Response · Authors · 2024-08-12
> > > **Adding Justifications/Explanations and Experiments**
> > >
> > > *Regarding the OOD application, my comment was more in line with reviewer L7r5, in that an experiment on real-world data (rather than a synthetic one) would significantly strengthen the paper.*
> > >
> > > **Response: Understood, thanks for the suggestion. We have now posted a "global" official comment above that discusses a real-world data experiment concerning credit card fraud detection. We are happy to include more real-world data experiments along these lines in the paper, if given the opportunity.**
> > >
> > > *Regarding the algorithmic-dependent results, a remark in the spirit of the last sentence of your response to this weakness  "Further, BH belongs to the family of step-up procedures, which means the rejection region is decided via a stopping time (see our analysis), which since our paper shows can be manipulated, it means other ``step-up" procedures" are similarly prone.) " would be worth including in a future version, perhaps along with more general theorem statements/corollaries.*
> > >
> > > **Response: Yes, we appreciate this suggestion and agree that this additional discussion surrounding the generality of attacking "step-up" procedures would be worthwile.**
> > >
> > > *My last remaining question for the author is whether they would be willing/able to add justifications/explanations in the paper in line with my and other reviewer's comments and questions on z-scores perturbation, choice of norm, adversary's power, etc. (which would make the paper clearer, a bit more solid and easier to approach for people who are not as familiar with the literature), and whether they have or will perform an experiment on real-world data as suggested by L7r5 and myself.*
> > >
> > > **Response: Yes, we are absolutely willing and able to add the justifications/explanations we offered to the review team on matters like z-score perturbation, norm choice, adversary's power, etc. And yes, we have performed one experiment on real-world data and offered it in the "global" official comment above, and we are willing to perform additional experiments on real-world data as suggested by L7r5 and yourself.**

---

### Author Rebuttal · Authors · 2024-08-07

We thank the reviewers for their questions and feedback. We have provided individual responses to each and welcome the opportunity to engage further in the discussion period.

---

> ### Comment · Area_Chair_NBc5 · 2024-08-12
>
> Dear Authors,
>
> There are still some pending clarification questions from at least two reviewers.  For example, would you commit to add experiments  on real-world data and explanations along the lines of what Reviewers L7r5 and pEvv request?
>
> We need your input.

---

> ### Author Response · Authors · 2024-08-12
> **On a Real-World data experiment (Credit Card Fraud Detection)**
>
> Thanks for letting us address the comments on real-world data. We have since conducted an experiment on the ``Credit Card" dataset, the same one used in the real-world data experiments in Bates, Candes, et al 2023. While we appreciate the suggestion to consider MNIST/FasionMNIST, we chose this experiment on this dataset, as it relates to fraud detection, which we feel may better align with the Safety in Machine Learning theme. We proceed to describe this experiment.
>
> The ``Credit Card" dataset contains 284,807 transactions transactions made by credit cards in September 2013 by European cardholders over the course of two days - 492 of which were frauds. With the exception of 'Time' and 'Amount', it contains numerical input variables that are the result of a PCA transformation. The 'Class' label takes value 1 in case of fraud and 0 otherwise.
>
> We trained an isolation forest (just as Bates et al 2023 did) on 141,758 randomly selected genuine transactions (that is, about 1/2 of the total) using the R library isotree. Then for each of $10^2$ simulations, we drew 141,657 genuine instances uniformly at random from the remaining 142,557 genuine transactions to form a calibration set, with the remaining 900 genuine transactions combined with a randomly drawn set of 100 forged instances drawn from the 492 forged collection to form a test set of 1000 credit card transactions. Running the score function on the test set members produced a collection of test set anomaly scores, which with the use of the calibration set, conformal p-values were generated. Hence, in each of the simulations that saw the random generation of calibration and test sets, a list of conformal p-values is generated, one for each corresponding test set credit card transaction. Next, we executed our adversarial INCREASE-c algorithm to generate an altered list $p$ of p-values, for varying $c$; in other words, $c$- many of the credit card transactions from the test set were altered to generate a slightly modified list $p_{+c}$. Then, we performed $BH_{0.1}$ twice, once on $p$, and once on $p_{+c}$.
>
> We report the average (over all simulations) false detection proportion (FDP) by $BH_{0.1}$ when there is no adversarial intervention, $E[FDP[BH_{.1}; p]]$, the average FDP incurred by $BH_{0.1}$ when the adversary implements INCREASE-c (for $c = 1, 5, 10, 20$), $E[FDP[BH_{.1}; p_{+c}]]$ , as well as the average rejection count $E[\tilde{k}]$ and the adversarially-adjusted rejection count $E[\tilde{k}_{+c}]$. See the summary table below.
>
> | $c$ |$E[FDP[BH_{.1}; p]]$ | $E[FDP[BH_{.1}; p_{+c}]]$ |$E[\tilde{k}]$ |$E[\tilde{k}_{+c}]$ |
> |-------|------------------------|----------------------------|-----------------|----------------------|
> | 1     | 0.09                   | 0.11                       | 62.6           | 64.27                 |
> | 5     | 0.09                   | 0.17                       | 48.69           | 57.12                 |
> | 10    | 0.09                   | 0.23                       | 56.39           | 72.85                 |
> | 20    | 0.09                   | 0.31                       | 58.12           | 89.06                 |
>
> The false detection proportion is the proportion of the total number claimed to be fraudulent that were in fact genuine. As we can see, although $BH_{0.10}$ can control the FDR below the explicit $0.10$ level, INCREASE-c's tampering can push the FDR above this control level.
>
> We are happy to conduct more real-world data experiments and add to the appendix, if given the opportunity to do so. We would be glad to share the relevant R code as well.

---

> > ### Comment · Reviewer_L7r5 · 2024-08-13
> >
> > Dear Authors,
> > Thank you for experimenting with the ``Credit Card" dataset. But I have one concern. If the experiments were conducted on a dataset containing both fraudulent and genuine transactions, what was the need to generate an altered list $p$ of p-values for varying $c$? Doesn't an adversary implicitly make the alterations in the fraudulent samples? Thus, changes in the fraudulent samples must have been mapped to the p-values and the z-scores. The authors should have first computed p-values and the z-scores from the input samples and then implemented the INCREASE-c algorithm to calculate its efficacy.
> >
> > Could the authors explain this? If the authors are again computing a perturbed set of p-values from the input samples, which are already manipulated by an adversary, then the feasibility of the proposed approach in real scenarios is lost.

---

> ### Author Response · Authors · 2024-08-13
> **Clarification on Credit Card Experiment and the role of the adversary for L7r5**
>
> *Question: Could the authors explain this? If the authors are again computing a perturbed set of p-values from the input samples, which are already manipulated by an adversary, then the feasibility of the proposed approach in real scenarios is lost.*
>
> **Response: The input samples are not "*already manipulated by an adversary*." If we are understanding correctly, you are under the impression that fraudulent credit card transactions are created by the adversary we are studying, which is not the case. Both genuine and fraudulent credit card transactions are phenomena that arise/occur independent of the adversary we are studying, albeit fraudulent ones are considered "anomalies" that we certainly wish to detect. And the adversary we are studying is acting to complicate this task of distinguishing the genuine from the fraudulent. More precisely, an anomaly detection system's task in this context is to help credit card companies distinguish fraudulent credit card transactions from genuine credit card transactions. Bates et al 2023 and other recent works would suggest developing an anomaly detection system that leverages BH's false discovery rate control so as to not make too many false positives which in this context translates to not mistakenly identify too many genuine transactions as fraudulent.**
>
> **In particular, Bates et al 2023 used exactly this same dataset to illustrate their methodology. They first trained an isolation forest on genuine transactions. Then they drew a random test set composed of both genuine and fraudulent transactions (the identity of which is effectively unknown to the decision-maker/AI system) in order to evaluate the capacity of their methodology to detect fraudulent transactions. Using the trained isolation forest along with a calibration set, they transformed the test set transactions into a corresponding collection of (conformal) p-values, for which they used BH to ultimately label each test set transaction as either genuine or fraudulent. While BH provides the guarantee that the expected false positive rate (or FDR) of this methodology can be kept below a user-defined threshold, we show that if an adversary were to step in and make (minimal) alterations to the test set of transactions, the methodology's FDR control could be broken. For details on this statement, we kindly refer you to the fact that our simulations indicated $E[FDP[BH_{0.1}; p]] \leq 0.10$, in line with the theory of BH; however, upon adversarial intervention on the test set, i.e., execution of INCREASE-c, we obtained instead $E[FDP[BH_{0.1}; p_{+c}]]$ that (significantly) exceeded 0.10. In other words, the adversary's corruption of $c-$ many test transactions could confuse the BH-based methodology into a greater FDR than it would claim, equiv., the average ratio of (the number of transactions falsely labeled as fraudulent) : (number of all transactions labeled as fraudulent) is higher than what would be guaranteed.**
>
> *Question: The authors should have first computed p-values and the z-scores from the input samples and then implemented the INCREASE-c algorithm to calculate its efficacy.*
>
> **Response: If we're understanding you correctly, this is in fact precisely what we did. It appears as though there has been a misunderstanding of what the Credit Card Data experiment is about. We sincerely hope our response in the preceding lines above have helped to clarify.**

---

### Decision · Program_Chairs · 2024-09-25

**Decision:**

Accept (poster)

**Comment:**

The paper investigates the conditions under which the Benjamini-Hochberg procedure is resilient to adversarial attacks.  In addition, the authors reframe the BH procedure as a "balls into bins" process as well as propose an algorithm to increase the rejection count. Finally, the authors also conducted experiments to support their findings.

All the reviewers had positive opinion on the paper, though in some cases this would come with low confidence as BH is not a well-known approach within the machine learning literature.  Nevertheless, good discussion was generated during the rebuttal period and on one hand several clarifications were provided or reviewers' questions were answered by the authors, and on the other hand the authors also agreed to add information on the final version of the paper that would make the final manuscript more accessible to everyone and on-par with other papers that are accepted in NeurIPS. Along these lines, in the final version of the paper clarifications are expected on matters like z-score, perturbation, norm choice, adversary power, etc.  Furthermore, the authors also performed experiments on a real-world dataset during the rebuttal period and these experimental results further elucidate the authors' approach as it was requested by the reviewers. I expect that these results will be included in the final version of the paper in one way or another.  For example, the authors may find it more reasonable to hint about these additional experimental results in the main text of their paper and provide full details in the supplementary material.